# Manic Fringe deficiency imposes Jagged1 addiction to intestinal tumor cells

Erika López-Arribillaga[1], Verónica Rodilla [1], Carlota Colomer[1], Anna Vert[1], Amy Shelton[2], Jason H. Cheng[2], Bing Yan[1], Abel Gonzalez-Perez[3], Melissa R. Junttila[4], Mar Iglesias[5,6], Ferran Torres[6], Joan Albanell[1,7], Alberto Villanueva[8], Anna Bigas[1], Christian W. Siebel[2] & LLuís Espinosa[1]

Delta ligands regulate Notch signaling in normal intestinal stem cells, while Jagged1 activates Notch in intestinal adenomas carrying active β-catenin. We used the $Apc^{Min/+}$ mouse model, tumor spheroid cultures, and patient-derived orthoxenografts to address this divergent ligand-dependent Notch function and its implication in disease. We found that intestinal-specific Jag1 deletion or antibody targeting Jag1 prevents tumor initiation in mice. Addiction to Jag1 is concomitant with the absence of Manic Fringe (MFNG) in adenoma cells, and its ectopic expression reverts Jag1 dependence. In 239 human colorectal cancer patient samples, MFNG imposes a negative correlation between Jag1 and Notch, being high Jag1 in the absence of MFNG predictive of poor prognosis. Jag1 antibody treatment reduces patient-derived tumor orthoxenograft growth without affecting normal intestinal mucosa. Our data provide an explanation to Jag1 dependence in cancer, and reveal that Jag1–Notch1 interference provides therapeutic benefit in a subset of colorectal cancer and FAP syndrome patients.

[1] Cancer Research Program. CIBERONC. Institut Mar d'Investigacions Mèdiques, Hospital del Mar, Doctor Aiguader 88, 08003 Barcelona, Spain. [2] Department of Discovery Oncology, Genentech, Inc., 1 DNA Way, South San Francisco, CA 94080, USA. [3] Research Program on Biomedical Informatics, IMIM Hospital del Mar Medical Research Institute and Universitat Pompeu Fabra, Doctor Aiguader 88, 08003 Barcelona, Catalonia, Spain. [4] Department of Translational Oncology, Genentech, Inc., 1 DNA Way, South San Francisco, CA 94080, USA. [5] Department of Pathology, Hospital del Mar, Doctor Aiguader 88, 08003 Barcelona, Spain. [6] Universitat Autonoma of Barcelona, Barcelona 08193, Spain. [7] Department of Oncology, Hospital del Mar, CIBERONC. Doctor Aiguader 88, 08003 Barcelona, Spain. [8] Laboratori de Recerca Translacional, IDIBELL-Institut Català d'Oncologia Gran Via km 2.7, Hospitalet, 08907 Barcelona, Spain. These authors contributed equally: Erika López-Arribillaga, Verónica Rodilla. Correspondence and requests for materials should be addressed to A.B. (email: abigas@imim.es) or to C.W.S. (email: siebel.christian@gene.com) or to L.E. (email: lespinosa@imim.es)

As the second leading cause of cancer-related deaths in Western populations, colorectal cancer (CRC) requires a deeper mechanistic understanding to fuel new therapeutic approaches. A series of well-characterized genetic alterations provide a framework to explain progression from adenoma to invasive CRC. In particular, human and mouse genetics identified inactivation of the adenomatous polyposis coli (*Apc*) tumor suppressor as an essential early event[1,2]. Apc loss activates Wnt/β-catenin that critically regulates intestinal stem cells (ISCs)[3,4] and tumor initiation[5]. Germline mutations in the *Apc* gene cause the inherited familial adenomatous polyposis (FAP) syndrome[6–9], which typically manifests with spontaneous polyp growth in the mid-teen years, progressing to more than a hundred adenomatous colon polyps by the age of 35. CRC ensues unless FAP patients undergo prophylactic colectomy. The best-characterized model for intestinal polyp formation is the $Apc^{Min/+}$ mouse. Like FAP patients, these mice develop early (4–5 weeks of age) multiple adenomas, mainly in the small intestine, with death ensuing after 4–8 months[10–12]. Progression from intestinal adenoma to advanced CRC, both in mice and humans, correlates with the accumulation of specific mutations and aberrant signaling through additional pathways[13].

The Notch pathway is an evolutionarily conserved signal for cell–cell communication, controlling lineage decisions in myriad tissues. Mammalian Notch signaling relies on four receptors (Notch1–4) and ligands of the Delta (Dll1, 3, and 4) or Jagged (Jag1 and 2) families. Receptor–ligand interactions are regulated by receptor glycosylation, with the Fringe family of N-acetylglucosaminyltransferases, mainly Lunatic Fringe (LFNG) and Manic Fringe (MFNG), potentiating Delta-induced signals and reducing responsiveness to Jagged ligands[14–17]. Interactions between Notch receptors and ligands, typically expressed on neighboring cells, trigger sequential proteolytic cleavages catalyzed by ADAM[18] proteases and the gamma-secretase complex[19]. The subsequently liberated intracellular domain of Notch (ICN) translocates to the nucleus and, together with RBP$_J$, activates transcription of target genes, often including members of the Hes family of transcriptional repressors[20,21]. Lineage studies in the mammalian intestine provide a paradigm for mammalian Notch signaling. Notch1 and Notch2, expressed on the crypt base columnar (CBC) stem cells, pair with Dll1 and Dll4, expressed on neighboring Paneth cells, to control cell proliferation and promote absorptive over secretory fates[22–24]. Pharmacological or genetic inhibition of Notch in the intestinal epithelium results in loss of the proliferative crypt compartment and replacement of intestinal stem and progenitors cells with post-mitotic secretory cells[25]. Notch not only induces expression of transcriptional repressors such as *Hes1* to prevent stem and progenitor cell differentiation, but also induces transcription of several ISC genes such as *Olfm4*[26] and *Bmi1*[27].

Collaborating Notch and Wnt signals control cell proliferation and fate in the normal crypt and have been linked to intestinal cancer, with Notch functioning downstream or in parallel with β-catenin[28,29]. Adenomas generated in *Apc* mutant mouse models display active Notch signaling and have been reported to be sensitive to gamma-secretase inhibitors (GSIs), which partially induce adenoma differentiation into post-mitotic goblet cells[25]. We previously identified Jag1–Notch signaling as an important link between Wnt and intestinal cancer, with *Jag1* as both a transcriptional target of β-catenin and a Notch signal inducer in *Apc*-mutant adenomas. Importantly, *Jag1* haploinsufficiency correlated with decreases in adenoma size and cell proliferation[29]. Differential usage of Notch ligands has been associated with expression of Fringe glycosyltransferases. In drosophila, Fringe was identified as a modifier of Notch receptor that favored the interaction with Delta and prevented the one with Serrate (Jag

ortholog)[30]. However, the molecular mechanisms supporting the differential use of Dll and Jag ligands in normal and transformed intestinal epithelial cells, respectively, remain completely unexplored. Using mouse genetic models, therapeutic antibodies, and organoid/tumor spheroid models, we now show that epithelial Jag1 is specifically required for tumor initiation, stem cell marker expression, and stem cell activity of the adenoma cells, both in vitro and in vivo. Furthermore, addiction of adenoma cells to Jag1 is mediated by MFNG. In agreement with this, JAG1 levels predict patient prognosis specifically in the group that carries MFNG-negative tumors.

## Results

**Intestinal epithelial Jag1 deletion blocks tumor initiation.** We previously reported that heterozygous deletion of Jag1 reduced adenoma cell proliferation and tumor size in $Apc^{Min/+}$ mice without affecting tumor number[29]. To determine whether intestine-specific *Jag1* deletion would be sufficient to affect adenoma growth or number, we functionally deleted both copies of *Jag1* in $Apc^{Min/+}$ mice by crossing the $Jag1^{lox/lox}$ strain[31] with *Villin*-Cre deleter mice[32]. A ROSA26-YFP reporter confirmed Cre activity in all intestinal epithelial cells, including CBC cells intermingled with Paneth cells[33] and the quiescent ISCs at the +4 position (Supplementary Fig. 1A). Although *Villin*-Cre is active as early as embryonic day 12.5[32], intestinal *Jag1* deletion did not affect growth (Supplementary Fig. 1B), impact intestinal integrity or alter the distribution of cell lineages (Supplementary Fig. 1C), consistent with observations in adult intestines following Jag1 deletion[34]. In contrast, in the $Apc^{Min/+}$ background, approximately four-fold fewer tumors arose following Jag1 homozygous deletion compared to Jag1 wild-type (WT) ($p = 0.0016$) or heterozygous deletion (HT) ($p = 0.038$), respectively (Fig. 1a, b). These results are different to those obtained using the general Jag1 heterozygous mice and indicate that epithelial Jag1 is required for tumor initiation. However, Jag1 expressed in the stroma of the tumor may also contribute to different aspects of tumor initiation and progression, as suggested from our previous work[29]. YFP expression (Supplementary Fig. 1D) and the absence of functional Jag1 mRNA in YFP+ cells from the rare tumors in $Jag1^{lox/lox}$ mice (Supplementary Fig. 1E) confirmed effective Jag1 exon 4 deletion. In the composite $Apc^{Min/+}$ mice, most Jag1 WT adenomas showed high levels of nuclear ICN1 (Fig. 1c) and total and nuclear Notch2 (Supplementary Fig. 1F) mainly restricted to the epithelial component, compared to the levels detected in tumors lacking *Jag1*. Further supporting the notion that Jag1 deletion inhibited Notch activity, we observed decreased expression of the Notch target genes *Hes1*, *c-Myc*, and *Nrarp* in *Jag1* knockout (KO) compared to WT tumors (Fig. 1d). Such decreased Notch activity correlated with reduced tumor cell proliferation, as assessed using Ki67 staining (Fig. 1c). However, we did not detect a significant increased in the number of cleaved-caspase 3-positive cells (apoptotic) in any Jag1 genotype (Supplementary Fig. 1G). In addition, $Apc^{Min/+}$ mice carrying the different *Jag1* genotypes showed a comparable distribution of dysplastic crypts, tubular and villous adenomas (Supplementary Fig. 1H), indicating that Jag1 requirement is not restricted to any particular tumor subtype.

Next, we exploited a potent and selective antibody antagonist targeting Jag1[35] to ask whether pharmacologic Jag1 inhibition would interfere with adenoma formation or growth in vivo. We began treating $Apc^{Min/+}$;*Villin*-Cre mice at 8 weeks of age, when some intestinal polyps have already been formed, with isotype control antibody or anti-Jag1 blocking antibody (Fig. 1e). After 10 weeks of weekly treatment, selective Jag1 blockade significantly reduced the number of polyps in the small intestine (SI) and

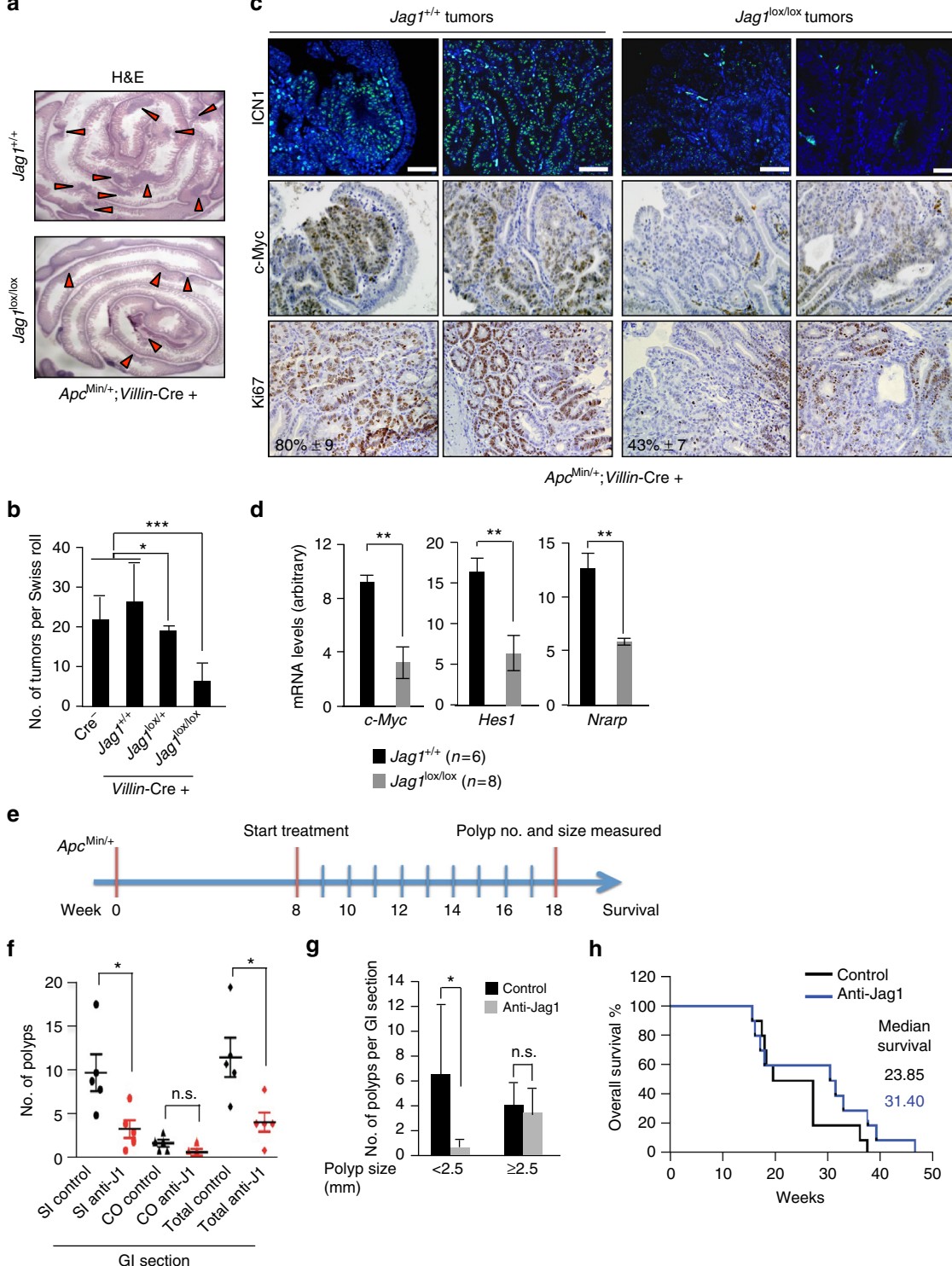

**Fig. 1** Jag1 is required for tumor initiation and tumor-associated Notch signaling. **a** Intestine Swiss-roll images from representative $Apc^{Min/+};Jag1$ wild-type ($Jag1^{+/+}$) or knockout ($Jag1^{lox/lox}$) mice obtained with the stereomicroscope. Red arrowheads denote intestinal tumors. **b** Quantification of tumor number per Swiss-roll section in the indicated genotypes analyzed at 2.5–3 months of age. **c** IF or IHC analysis of two representative $Apc^{Min/+};Jag1^{+/+}$ or $Jag1^{lox/lox}$ adenomas showing the levels of active Notch1 (ICN1) and c-Myc, and the proportion of proliferating cells determined by Ki67 staining. **d** qRT-PCR analysis to determine the mRNA levels of the indicated Notch targets in tumors from different genotypes relative to $\beta2$-microglobulin. **e** $Apc^{Min/+};Villin$-Cre mice were dosed weekly for 10 weeks with 10 mg per kg of either isotype control (black in **f**) or anti-Jag1 blocking antibodies (red in **f**) starting at 8 weeks of age ($n = 5$ per treatment group), near the onset of polyp formation. **f, g** Number of polyps that grew in total ($p = 0.0175$), in the small intestine (SI) ($p = 0.025$), where most polyps arise in this model, or in the colon (CO) (**f**) and analysis of polyp size in the same experiment (**g**). **h** Overall survival analysis in separate cohorts of 10 animals per treatment group that received weekly doses beginning at 8 weeks of age. Bars represent mean values ± standard error of the mean (s.e.m.); $p$ values were derived from an unpaired $t$-test, two-tailed (*$p < 0.05$, **$p < 0.01$, and ***$p < 0.001$). Scale bars represent 75 µm

almost precluded the formation of colonic (CO) tumors (Fig. 1f), with no detectable signs of systemic toxicity. Intriguingly, in anti-Jag1-treated animals, small tumors (< 2.5 mm) were essentially undetectable (Fig. 1g), while no difference was observed with larger tumors. These data are consistent with our genetic results, bolstering the notion that antibody treatment completely precludes polyp initiation induced by $Apc$ mutations. In contrast, it did not affect the growth of well-established polyps present at the start of treatment. Importantly enough, overall survival trended longer following anti-Jag1 treatment (Fig. 1h), although the effect was not statistically significant.

Together, these results indicate that $Jag1$ is not essential for normal intestinal homeostasis, but it is required for β-catenin-driven adenoma formation. Thus, genetic deletion of epithelial Jag1 or systemic inhibition using a blocking antibody is sufficient to prevent adenoma formation in the $Apc^{Min/+}$ mice, with no evident signs of toxicity.

**$Jag1$-deficient adenomas reduce stem cell gene transcription**. To explore the mechanism by which $Jag1$ deletion or inhibition impacts adenoma formation, we compared the expression levels of several stem cell markers in the $Jag1$ KO versus WT adenomas. Supporting a role for Jag1 in adenoma formation and by extension in tumor-initiating cell (TIC) activity, $Jag1$ KO adenomas expressed reduced levels of $Lgr5$, $Bmi1$, $Ephb2$, $c$-$Myc$, $Sox9$, and $Cd133$ compared to the $Jag1$ WT, as determined by qRT-PCR (Fig. 2a) and in situ hybridization (ISH) (Fig. 2b, c). Immuno-fluorescence (IF) analysis of endogenous LGR5 protein confirmed that this particular stem cell-like tumor population was reduced in the $Jag1$ KO mice (from 48 in the WT to 21% in average), whereas the ISC compartment of normal crypts remained unaffected (Fig. 2d). We also determined the expression levels of the intestinal differentiation markers in these adenomas. We found that presence of Paneth cell in the tumors, as determined by lysozyme (Lyz1) detection, negatively correlated with the number of functional Jag1 alleles, being KO tumors more frequently Lyz1 positive (70%) compared to HT (40%, $p < 0.05$) or WT (20%, $p < 0.01$) tumors (Fig. 2e). In contrast, expression of the absorptive marker carbonic anhydrase II (CAII) was virtually lost in the Jag1 KO tumors ($p < 0.01$, compared to the WT tumors). These results suggest that Notch signaling through Jag1 promotes stemness specifically in the intestinal adenomas while preventing differentiation toward the secretory lineage.

**Jag1 is required for tumor spheroid growth in vitro**. We functionally assayed whether Jag1-induced Notch signaling regulates TIC activity of intestinal adenomas by using the well-established model of 3D cultures that allows the maintenance of tissue-specific stem cells. In particular, we isolated $Jag1$ WT or KO adenoma cells from ROSA26-YFP;$Apc^{Min/+}$ mice (Fig. 3a) and seeded under standard culture conditions for organoid and tumor/adenoma spheroids[24,36–39]. In three independent experiments, we obtained 70–80 adenoma spheroids per well from 50,000 isolated YFP+ $Jag1$ WT cells seeded, in contrast with the $Jag1$ KO adenoma cells that completely failed to form spheroids (Fig. 3b). As expected, $Jag1$ WT spheroids grew as secondary spheroids and after subsequent passages with comparable potential (not depicted), reflecting their stem cell activity. Treatment with the GSI DAPT inhibited growth of primary and secondary spheroids (Supplementary Fig. 2A), consistent with growth depending on Notch signaling. $Apc^{Min/+}$ adenoma cells growing as three-dimensional (3D) structures displayed high protein levels of the ISC markers $Lgr5$, $Bmi1$, $Cd44$, and $Ephb2$, as well as Jag1 with rare cells expressing the intestinal differentiation markers Muc2, synaptophysin, CAII, and lysozyme

(Supplementary Fig. 2B), in contrast to the non-transformed organoid structures (Supplementary Fig. 2C). To further study the mechanisms underlying $Jag1$ dependence in this model, we generated the composite murine line $Apc^{Min/+}$;Jag1$^{lox/lox}$;β-$actin$-Cre-ERT, in which deletion of exon 4 of $Jag1$ can be induced with 4-OH tamoxifen resulting in a non-functional JAG1 protein. In the absence of 4-OH tamoxifen, spheroids derived from these adenoma cells grew similarly to those derived from the standard $Apc^{Min/+}$ strain. Treatment with 5 μM 4-OH tamoxifen prevented spheroid growth when supplied at the beginning of the culture (not depicted), and eradicated preformed $Apc^{Min/+}$;Jag1$^{lox/lox}$;β-$actin$-Cre-ERT spheroids after 3–4 days of treatment (Fig. 3c). In contrast, 4-OH tamoxifen treatment did not affect Jag1$^{lox/+}$ (Fig. 3c) or Jag1$^{+/+}$ spheroids (not depicted). Treatment with 4-OH tamoxifen for 24–48 h decreased ICN1 levels even in the presence of Dll4 (Supplementary Fig. 2D), and led to a reduction in the mRNA levels of the Notch targets $Hes1$, $Nrarp$, and $c$-$Myc$, and the stem cell markers $Ephb2$, $Bmi1$, $Hopx$, $Lrig1$, and $mTert$ (Fig. 3d). By IF analysis, we found that 4-OH tamoxifen treatment diminished the number of proliferating Ki67-positive cells and increased the quantity of lysozyme-(Paneth) and Alcian Blue-positive (mucosecretory) cells without significantly affecting CAII levels (Fig. 3e–g). These changes correlated with an increase in the number of apoptotic cells, as determined by active-caspase 3 staining at 48 h after 4-OH tamoxifen treatment closely resembling the effects of Notch inhibition by DAPT (Fig. 3e). Interestingly, active-caspase 3 and Alcian blue stainings were both detected in cells delivered into the spheroid lumen suggesting that Notch inactivation following Jag1 deletion imposes the terminal mucosecretory differentiation of adenoma cells leading to cellular apoptosis. Similarly, treatment of $Apc^{Min/+}$ spheroids for 72 h with the antibody targeting Jag1 reduced cell viability when compared with the antibody targeting Dll4. Remaining Jag1-treated spheroids displayed smaller and less complex glands, which contain pleomorphic cells with bigger nuclei and less cytoplasm (Fig. 3h).

**Absence of MFNG provides Notch ligand specificity**. The observation that adenoma initiation in the $Apc^{Min/+}$ intestines required Jag1-induced Notch signaling suggested that either Jag1 was the only Notch ligand expressed in the adenomas, or adenoma-specific Notch1 was refractory to the induction by other ligands such as Dll4. IF staining revealed that Jag1 and Dll4 were present at comparable levels, and localized as expected at the membrane of both normal organoids and adenoma-derived spheroids (Fig. 4a). To explore the possibility that differential expression of Fringe glycosyltransferase enzymes could explain a shift from Dll- to Jag1-induced Notch signaling[40], we examined the protein levels of LFNG and MFNG by IF. We did not detect any LFNG staining in our 3D intestinal cultures (not depicted). In contrast, MFNG was detected in most of the non-transformed organoid structures and notably absent in the $Apc^{Min/+}$ adenoma spheroids (Fig. 4a). IF examination of frozen (Supplementary Fig. 3A) and paraffin-embedded $Apc^{Min/+}$ intestinal sections (Fig. 4b), using two different anti-MFNG antibodies, confirmed that MFNG levels were significantly reduced in the adenoma tissue, in clear contrast to the consistent staining observed in adjacent normal crypts and at the base of the tumor area. MNFG levels were also increased in tumors arising in the $Apc^{Min/+}$ Jag1 KO intestines (Supplementary Fig. 3B) further suggesting that MNFG levels regulate addition to Jag1. In the same direction, non-transformed organoids carrying the $Jag1^{lox/lox}$; β-$actin$-Cre-ERT alleles, which express MFNG (not depicted), were totally refractory to 4-OH-tamoxifen treatment (Fig. 4c, upper panels). Efficiency of Jag1 (exon 4) deletion in the organoids

following 4-OH-tamoxifen treatment was confirmed by PCR (Fig. 4c, lower panel).

To experimentally test whether ectopic MFNG reverted JAG1 addiction in the adenoma cells, we transduced $Apc^{Min/+}$;

Jag1$^{lox/lox}$;β-*actin*-Cre-ERT spheroids with retroviral control (pMIG) or a vector codifying for MFNG, and then treated the cultures with 5 μM 4-OH-tamoxifen to deplete adenoma cells from functional JAG1. We found that Jag1$^{lox/lox}$ spheroids

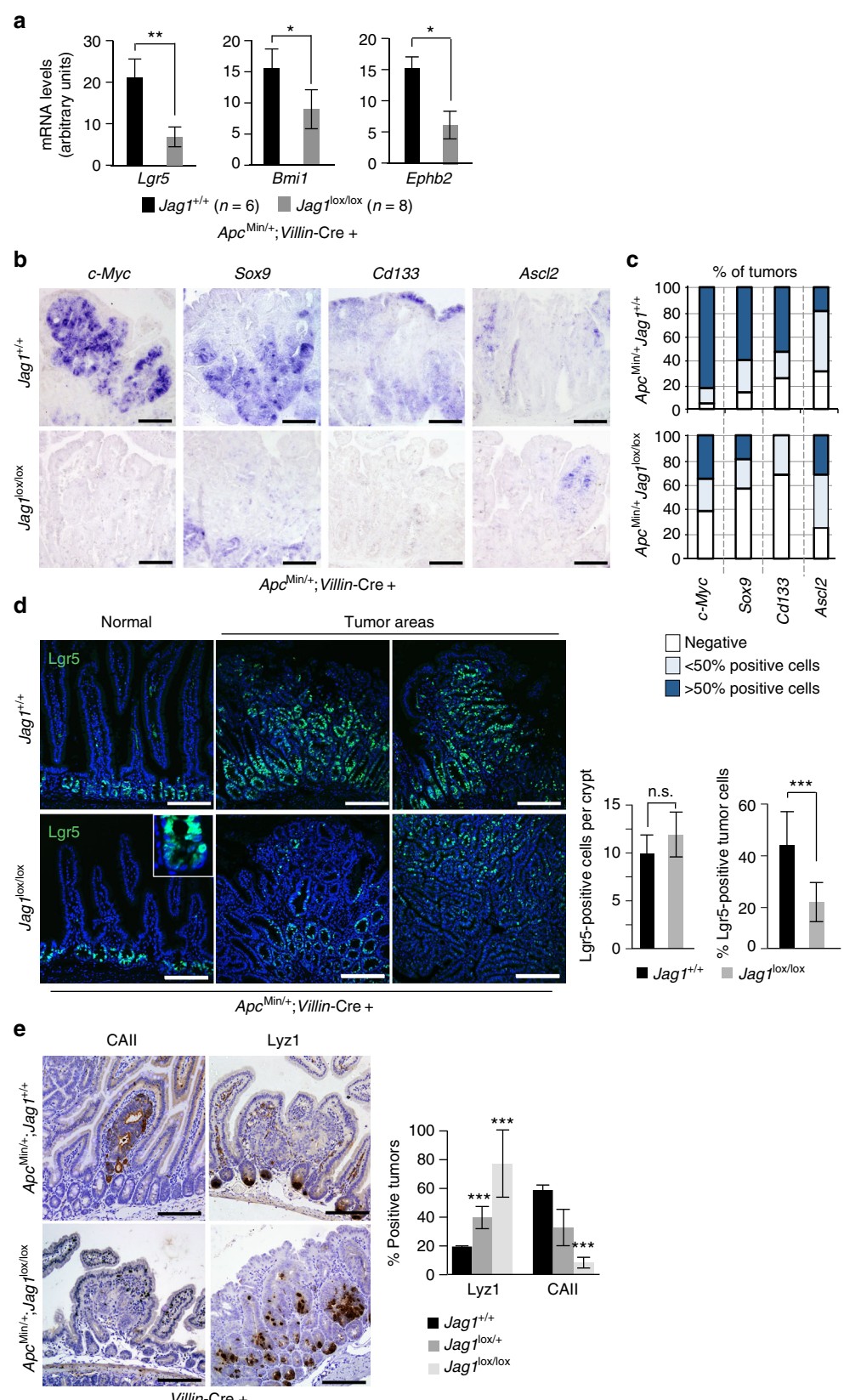

expressing ectopic MFNG were morphologically indistinguishable from the controls, although they were significantly protected from the death induced by 4-OH-tamoxifen (Fig. 4d, e).

**MFNG and Jagged1 expression predicts poor prognosis**. We then studied whether our results applied to human CRC. We performed immunohistochemistry (IHC) analysis of Jag1 and active ICN1 (Fig. 5a) in a cohort of 135 human adenomas (Supplementary Table 1) and 242 carcinoma samples (Supplementary Table 2). We found a significant correlation between the levels of Jag1 and ICN1 in the adenoma samples (58 of 73 Jag1 high were ICN1 high; 80%, $p = 0.0015$) that was lost in the carcinomas (Supplementary Fig. 4A). Since ability of Notch to respond to Jagged or Delta ligands in the adenomas is mostly defined by MFNG expression, as we have found in the mouse system, we reasoned that differences on MFNG levels might explain these conflicting results. IF analysis of MFNG in our set of samples (Fig. 5a and Supplementary Table 2) demonstrated that MFNG levels were significantly divergent in the adenoma and carcinoma samples ($p < 0.0001$) with MFNG primary lost in the adenomas (46% of MFNG-negative adenomas compared with 17% of the carcinomas) (Supplementary Fig. 4B). More refined analysis of ICN1 and Jag1 levels in the MFNG-positive and -negative carcinoma samples demonstrated an inverse correlation between Jag1 and ICN1 in the MFNG-positive group ($p = 0.037$) whereas ICN1 detection was biased toward the JAG1 high group in the absence of MFNG (nine out of 14 MFNG-negative and ICN1-high tumors were JAG1 high) (Supplementary Table 2). Importantly, we further analyzed Dll4 levels in a subset of CRC samples and found that ten out of 11 tumors that were characterized as ICN1 high/JAG1 low contained high levels of Dll4, nine of them being MFNG positive (Supplementary Table 3), further supporting that tumors expressing MFNG may induce Notch1 through Dll4. Clinically relevant, inside the group of patients carrying ICN1 high tumors, stratification according to JAG1 and MFNG was predictive of disease outcome, being the worst prognosis group that defined by JAG1 high and MFNG negative ($p = 0.0148$, compared with the JAG1 low/MFNG positive, and $p = 0.04$ compared with the unsorted population) (Fig. 5b). In contrast, we did not observe any association between JAG1 and/or MFNG levels and patient outcome in the Notch-negative population (Supplementary Fig. 4C). In a public cohort of 359 CRC tumors with both expression and clinical data available (probed by TCGA[41]), patients in the Notch high group with high MFNG expression do have better survival compared with the low MFNG group, although differences did not achieve statistical significance (Fig. 5c). Levels of MFNG in this subgroup of tumors also determined the transcriptional status of several Notch target genes such as NKD1, DLX3, EREG, EPHA4, or IL7R (Fig. 5d), known to be relevant for cancer progression[42–46]. Notably, MFNG levels were significantly lower in the tumors included in the CRC subtype 2[47] (Supplementary Fig. 4D), which is the most frequent and is characterized by a marked activation of the WNT and MYC pathways.

**Jag1 blocking antibodies reduce CRC growth in vivo**. Finally, we investigated the possibility that anti-Jag1 antibodies provide a therapeutic benefit for human colorectal tumors. From a library of primary human CRC developed as orthoxenografts in nude mice, we selected a tumor that efficiently grew in 3D cultures and lacked MFNG expression (Fig. 6a). A fragment of the tumor was expanded subcutaneously in nude mice and after that, comparable tumor fragments were orthotopically implanted in the cecum of 12 mice. When tumors were detectable by palpation, mice were randomly separated in two groups of treatment. Animals were treated twice a week by intraperitoneal injection with anti-Jag1 ($n = 6$) or isotypic control antibody ($n = 6$), and 40 days after initiating the treatments, animals were euthanized and analyzed for the presence of tumors, either in the intestine or associated to the wall of the peritoneum (scheme in Fig. 6b). Treatment of the animals with the anti-Jag1 antibody significantly reduced the size and weight of implanted tumors (Fig. 6c, d and Supplementary Figure 5A), although we did not observed a significant change in the number of peritoneal implants between both groups (three animals with implants in the control group and two animals in the treated group). Importantly, IHC analysis of the tumors demonstrated that anti-Jag1 treatment imposed a significant increase in the extent of necrosis and fibrosis inside the tumor areas and a reduction in proliferation in the residual tumor mass as determined by Ki67 staining (Fig. 6e, f), which was not observed in the non-transformed adjacent colonic tissue (Fig. 6g). Further indicating that human CRC are highly addicted to Notch1 signaling, we still detected ICN1 staining in the few growing areas of anti-Jag1-treated tumors (Fig. 6h). Using an in vitro system of patient-derived tumoroids ($n = 5$), we confirmed that anti-Jag1 treatment was more effective in tumors carrying high levels of JAG1 and low MFNG such as PDOXT005, PDOXT007, and PDOXT008 (Supplementary Fig. 5B).

Together, our findings demonstrate that mouse adenomas and a particular subset of human CRC tumors require Jag1-mediated Notch signaling for effective growing and to escape from apoptosis. In this context, selective anti-Jag1 blocking antibodies can be considered as plausible therapy for FAP disease but also for the treatment of a selected group of CRC patients (see model in Fig. 7).

## Discussion

Strategies for therapeutically targeting Notch signaling in CRC have suffered from numerous challenges. Clinical and pre-clinical evaluations of GSIs have uncovered toxicities, including intestinal goblet cell metaplasia associated with ISC loss. While intermittent dosing or co-treatment with glucocorticoids may open a therapeutic window[48], selective antibody inhibitors of individual Notch ligands or receptors provide an alternative therapeutic strategy. Indeed, compared to dual inhibition of Notch1 and Notch2, selective inhibition of either receptor alone minimized intestinal toxicity[22], consistent with functional redundancy of these two receptors in intestinal homeostasis[23]. Unfortunately,

**Fig. 2** Jag1 deficiency results in reduced levels of ISC markers in the tumor cells. **a** qRT-PCR to measure the mRNA levels of selected ISC markers in wild-type and Jag1-deficient adenomas. The average and standard deviation (s.d.) of five tumors analyzed is represented. **b** In situ hybridization to determine the expression pattern of c-Myc and different stem cell markers in tumor sections from the indicated genotypes. **c** Quantification of the percentage of tumors containing detectable levels of the indicated genes as determined by ISH. A minimum of 20 tumors of each genotype was included in the analysis. **d** IF analysis of LGR5 protein in normal or adenoma tissue in the indicated genotypes and quantification of 15 randomly selected areas (right panels). **e** IHC analysis of the Paneth cell marker lysozyme and the absorptive marker carbonic anhydrase II (CAII) in tumors from the indicated genotypes. Graphs represent the number of tumors expressing these markers in the indicated genotypes. Unpaired t-test, two-tailed, was used to determine the statistical significance of the differences (***$p < 0.001$). Scale bars represent 75 μm

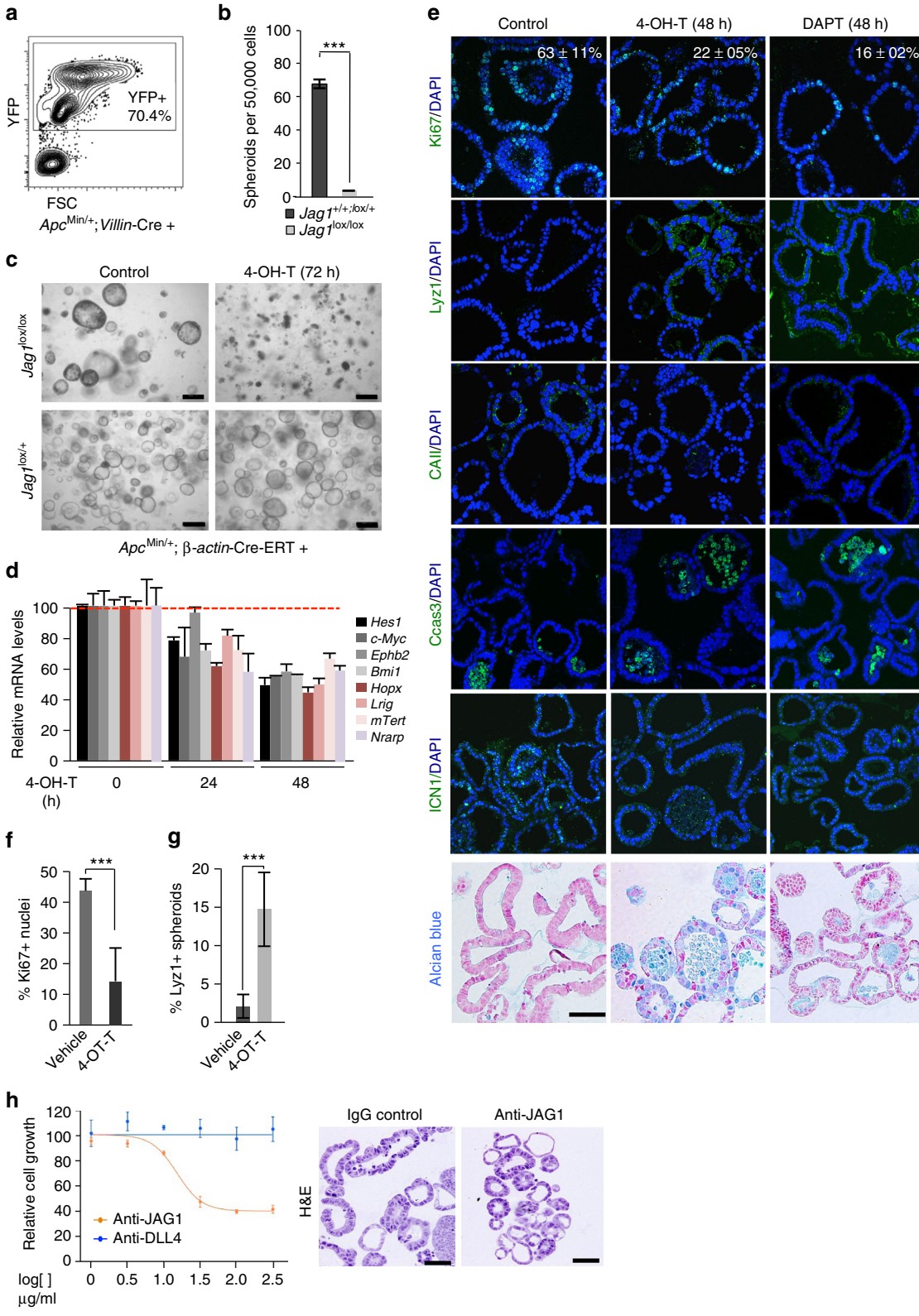

such redundancy may be maintained in CRC[29]. Our discovery of Jag1 dependence in $Apc^{Min/+}$ adenomas raises the possibility of an alternative strategy. We have found that intestinal epithelial Jag1 deletion prevents $Apc^{Min/+}$ tumor formation, reduces expression of several ISC markers, and inhibits tumor spheroid growth and re-passaging, a measure of TIC activity. In contrast,

Jag1 is dispensable for intestinal homeostasis, and systemic JAG1 inhibition does not cause goblet cell metaplasia. This is revolutionary, since we could treat patients with JAG1-blocking antibody targeting exclusively the tumor cells and avoiding the strong sides effects in the normal gut of FAP patients. Multiple lines of evidence point to a TIC population, which shares such a

**Fig. 3** Epithelial Jag1 is required for intestinal adenoma cells to grow in 3D cultures. **a** Representative FACS plot showing the sorted YFP⁺; $Apc^{Min/+}$ adenoma cell population used in the spheroid cultures. **b** Sorted $Apc^{Min/+};Jag1^{+/+};Villin$-Cre⁺ or $Apc^{Min/+};Jag1^{lox/lox};Villin$-Cre⁺ intestinal adenomas cells were seeded in the spheroid culture conditions. Graph shows the quantification from three biological replicates after 7 days in culture. **c** Representative images of $Jag1^{lox/+}$ or $Jag1^{lox/lox};\beta$-actin-Cre-ERT spheroids left untreated or treated for 4 days with 5 μM 4-OH-tamoxifen. **d** qRT-PCR analysis to determine the expression levels of the indicated genes in $Jag1^{lox/lox};\beta$-actin-Cre-ERT spheroids at 0, 24, and 48 h of 5 μM tamoxifen (OH-T) treatment. Values were normalized to the average value of β-2 microglobulin and Gapdh, and represented as percent of the vehicle-treated cultures. **e** IF analysis of the indicated proteins and alcian blue staining in spheroid cultures embedded in paraffin following the indicated treatments (OH-T 5 μM, DAPT 2.5 μM). **f**, **g** Graphs represent the quantification of the analysis shown in **e**. The average values from three biological replicates and standard deviation (s.d.) of the mean are shown. In all experiments, bars represent mean values ± standard deviation of the mean; p values were derived from an unpaired t-test, two-tailed. **h** Survival curves of $Apc^{Min/+}$ spheroid cultures treated with the blocking anti-Jag1 or anti-DLL4 antibodies (left panel) and representative hematoxylin and eosin (H&E) staining of anti-Jag1-treated spheroids (right panel). Scale bars represent 200 μm in (**c**) and 50 μm in (**e**) and (**h**)

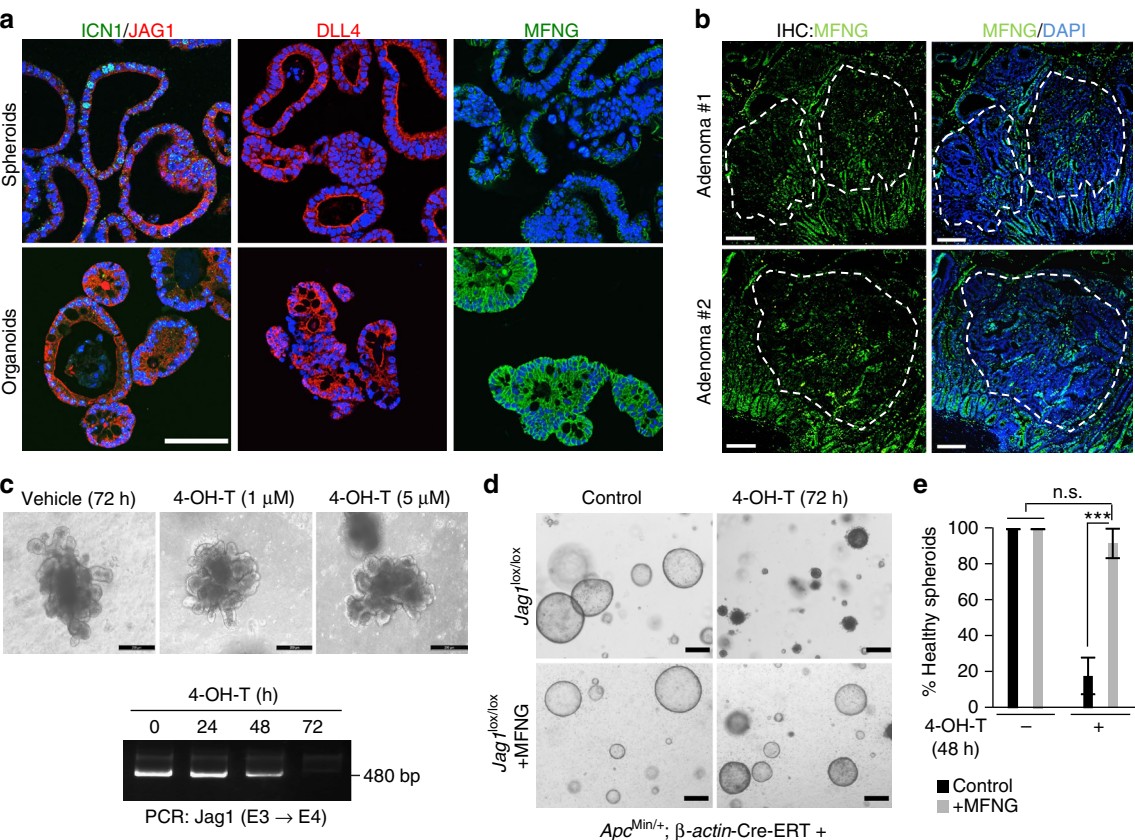

**Fig. 4** Reduced levels of MFNG in intestinal adenomas impose Notch-ligand specificity. **a** IF analysis of the indicated proteins in organoid and spheroid cultures embedded in paraffin. **b** IF analysis of MFNG in paraffin sections of two different $Apc^{Min/+}$ intestinal adenomas. Representative images are shown. **c** Representative images of non-transformed $Jag1^{lox/lox};\beta$-actin-Cre-ERT organoids left untreated or treated for 3 days with the indicated doses of 4-OH-tamoxifen. RT-PCR of organoids treated as indicated showing the functional wild-type band, which is deleted upon treatment. **d**, **e** $Jag1^{lox/lox};\beta$-actin-Cre-ERT adenoma cells were infected with a retroviral vector codifying for MFNG, then grown in matrigel to generate tumor spheroids, and treated with 4-OH-tamoxifen for 48–72 h. Representative images (**d**) and quantification of six individual wells counted per condition (**e**). Bars represent mean values ± standard error of the mean (s.e.m) from two independent experiments performed; p values were derived from an unpaired t-test, two-tailed. Scale bars represent 50 μm in (**a**), 100 μm in (**b**), 75 μm in (**c**), and 200 μm in (**d**)

transcriptional program and other biological features with normal ISCs[49], but is responsible for tumor relapse, metastasis, and chemotherapy resistance[50–52]. Our data suggest that β-catenin-driven intestinal TIC population rely on JAG1.

When we then looked at human CRC, we found a mixed pattern for JAG1 expression and no correlation between JAG1 and ICN1 levels. However, reduced MNFG levels determine a distinctive addiction to JAG1 and might be valuable in stratifying patient prognoses. Importantly, lower tumor expression of MFNG significantly associated with poor CRC prognosis specifically in the group of JAG1-high tumors. We propose that JAG1-high/MFNG-low highlight a CRC subset that could benefit from JAG1 therapeutic inhibitors. For example, MNFG is poorly expressed in the CRC subtype 2, which is characterized by marked activation of the WNT and MYC pathways. Taken together, our findings provide a framework for considering a Jag1-selective blocking antibody as a therapeutic candidate in subsets of CRC patients, particularly those carrying ICN1-high/MFNG-low tumors as well as those carrying germline $Apc$ mutations[53]. In fact, this possibility has been experimentally validated in a particular human CRC obtained from our orthoxenografts bank.

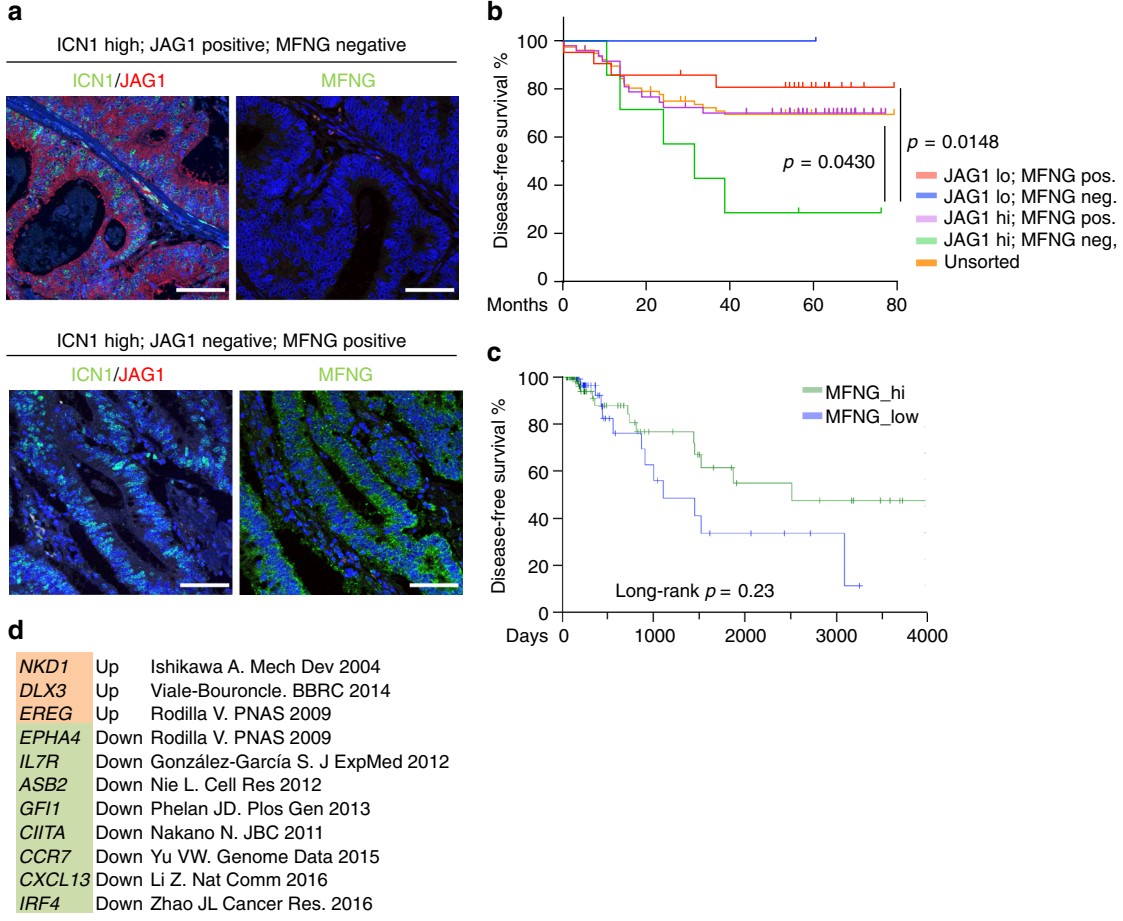

**Fig. 5** ICN1, JAG1, and MFNG levels predict CRC patient prognosis. **a** Representative images of Notch, JAG1, and MFNG staining in two different carcinoma samples. Scale bars represent 75 μm. **b**, **c** Disease-free survival analyses (Kaplan–Meier curves) of the indicated groups of patients determined by IF analysis of our cohort of patients (**b**) or from the TCGA CRC tumors in the subgroup (**c**) with high Notch1 expression that exhibit high (green) or low (blue) expression of MFNG. **d** Genes previously identified as Notch targets in the indicated publications that were up- (Up) or downregulated (Down) in the MFNG-low tumors compared to the MFNG-high

## Methods

**Animals**. $Jag1^{lox}$ mice[31] with lox sites flanking exon 4 (that codifies for the DSL domain) were crossed with Lox-STOP-lox-ROSA26YFP (YFP$^{lox}$)[54] and *Villin*-Cre and then with the $Apc^{Min/+}$ mice (both from Jackson Laboratories). Mice were genotyped by PCR and kept under pathogen-free conditions. The Animal Care Committee of Generalitat de Catalunya and the Genentech Institutional Animal Care and Use Committee (IACUC) approved all animal studies.

**Antibodies**. In the different analysis, we used primary antibodies against lysozyme (1:5000, Dako A0099); GFP/YFP (1:200, Takara 632460); Ki67 (1:500, Novocastra MM1); Myc (1:100, Santa Cruz sc-764); cleaved-caspase 3 (1:500, Cell Signaling #9661); MFNG (1:100, Santa Cruz sc-8237, Ab#1 and Biorbyt orb157845, Ab#2); Jagged1 (1:500, Santa Cruz sc-6011); Delta4 (1:1000, Abcam Ab7280); Muc2 (1:400[55]), CAII (1:2000, Rockland 200-401-136), Notch2 (1:500, Ab118824), and active ICN1 (1:200, Cell Signaling #4147). Lgr5 (GPCR49, Ab75732). Blocking anti-Dll4 antibody was provided by Genentech.

**ISH**. Murine intestinal samples were flushed gently with cold phosphate-buffered saline and fixed overnight in 4% paraformaldehyde at room temperature. Samples were then dehydrated, embedded in paraffin, and sectioned at 8 μm. After de-paraffinization and rehydration, the samples were treated with 0.2 N HCl and proteinase K (30 μg/mL). Samples were hybridized for 24 h at 65 °C. After washing, blocking solution was added (Blocking reagent, Roche) and incubation with anti-digoxigenin antibody was performed overnight at 4 °C. Next morning, samples were washed and developed with NBT/BCIP (Roche). The RNA probes were obtained from complementary DNA of mouse *c-Myc*, *Sox9*, *Cd133*, and *Ascl2* and were generated by in vitro transcription with a Digoxigenin RNA Labeling Kit (Roche) according to the manufacturer's instructions.

**qRT-PCR analysis**. Total RNA was extracted with the RNeasy Mini Kit (Qiagen) and the RT-First Strand cDNA Synthesis Kit (Amersham Pharmacia Biotech, GE Healthcare, Buckinghamshire, UK) was used to produce cDNA. qRT–PCR was performed in LightCycler480 system using SYBR Green I Master Kit (Roche, Basel, Switzerland). Primers used are detailed in Supplementary Table 4.

**Mouse adenoma dissociation and cell sorting**. Cell dissociation was performed as previously described[36,37]. Briefly, pooled adenomas were incubated in 8 mM EDTA for 20 min at 4 °C and the remaining un-soluble fraction treated with 0.4 mg/mL dispase and subsequently with 1.25 mg/mL collagenase 20 min each at 37 °C in agitation. Cell suspension was recovered by centrifugation at 1200 rpm and suspended in 140 nM ROCK inhibitor (Y-27632, Sigma). We determined cell viability by Trypan Blue dye exclusion or DAPI staining (VYSIS). Viable cells were used for either direct cell culture, or sorted (based on YFP expression) in a FACSAriaII (BD Biosciences) and collected in HBSS with antibiotics and 140 nm ROCK inhibitor.

**Mouse intestinal organoid and tumor spheroid cultures and staining**. Approximately $10–20 \times 10^4$ cells from either normal intestinal murine crypts (organoids) or dissociated intestinal adenomas (spheroids) from $Apc^{Min/+}$ mice were seeded in 50 μL Matrigel (BD Biosciences) in 24-well plates. After polymerization, 500 μL of complete spheroid medium [DMEM/F12 plus penicillin (100 U/mL) and streptomycin (100 μg/mL) (Biological Industries); N2 and B27 (Invitrogen); 140 nM ROCK inhibitor, 100 ng/mL Noggin, and 20 ng/mL bFGF (Peprotech); 100 ng/mL R-spondin (R&D Systems) and 50 ng/mL EGF (Sigma)] was added. Cultures were maintained at 37 °C, 5% $CO_2$ and medium changed every 2 days. According to standard protocols, we also added Wnt3 (R&D Systems) at the start of the organoids cultures to facilitate their formation, and then removed it. Notch inhibitor DAPT (Calbiochem) was used at 50 μM. Inducible deletion of Jag1

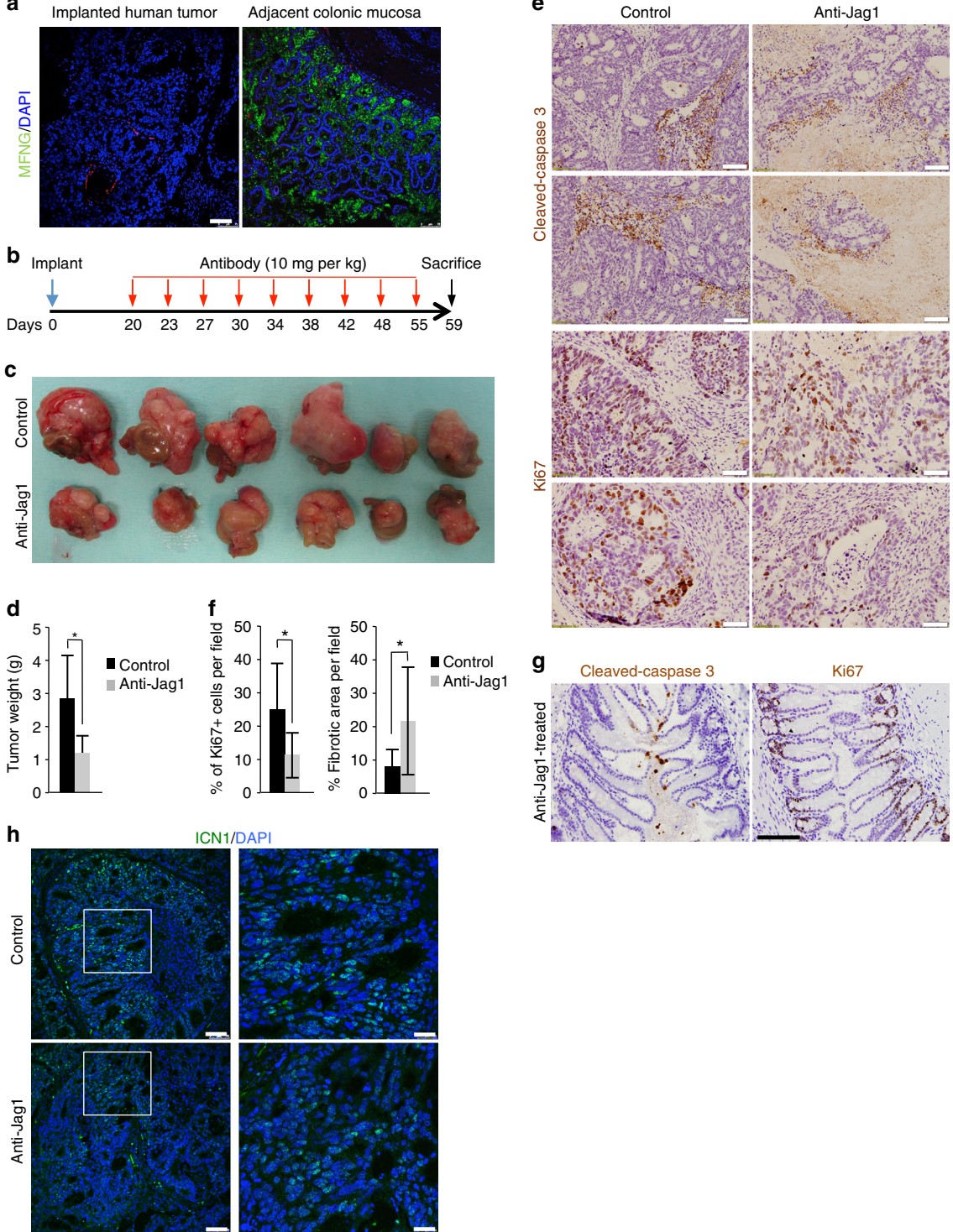

**Fig. 6** Treatment with antibody against Jag1 decreases human CRC tumor growth in vivo. **a** IF analysis of MFNG expression in the transplanted human tumor and the adjacent normal colonic mucosa. **b** Scheme of the protocol used for testing the anti-tumor activity of anti-Jag1 antibody in vivo (10 mg per kg). **c** Photograph of the tumors recovered at the end of the experiment with the indicated treatments. **d** Median and standard deviation of tumor weight in the animals from both groups of treatment. **e** Representative images of IHC analysis of cleaved-caspase 3 and the proliferation marker Ki67 in two representative tumor sections from control and anti-jag1-treated animals and **f** average values and standard deviation of the data obtained from eight independent 20× fields analyzed using the *ImageJ* program. Statistical significance of the differences between control and anti-Jag1-treated groups was determined by *t*-test comparing. **g** IHC analysis with the indicated antibodies in normal adjacent tissue. **h** IF analysis of active Notch1 (ICN1) in representative control and anti-Jag1-treated tumors. Scale bars represent 75 μm in (**a**), 100 μm (cleaved-caspase 3) and 50 μm (Ki67) in (**e**), 200 μm in (**g**), and 75 μm and 25 μm (magnification) in (**h**)

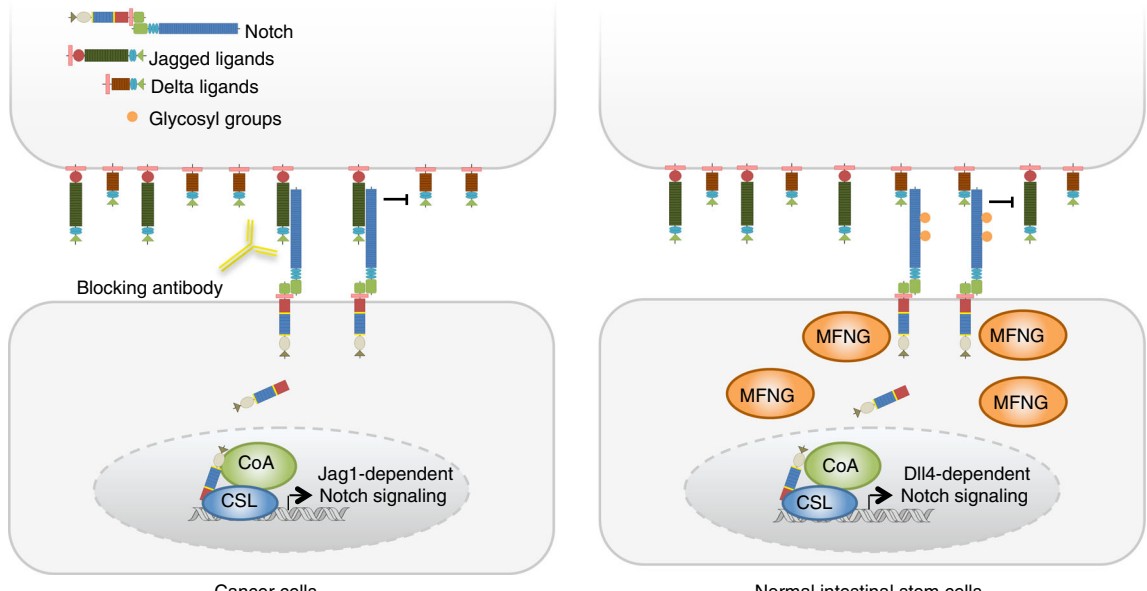

**Fig. 7** Model representing the differential requirement for Jag1 in the normal and transformed intestinal epithelial cells. As shown in the figure, MFNG in normal intestinal stem cells glycosylate Notch1 thus favoring its activation mediated by the Dll4 ligand that is localized in the adjacent (Paneth) cells. In contrast, non-glycosylated Notch1 present in the MFNG-negative tumor cells cannot respond to Dll4 leading to Jag1, which makes them addicted to Jag1-dependent Notch signaling

in the preformed $Apc^{Min/+}$;$Jag1^{lox/lox}$;β-actin-Cre-ERT spheroids was achieved by treating the cultures with the indicated doses of 4-hydroxytamoxifen up to 72 h.

For whole-mount immunostaining analysis, organoids and spheroids were fixed with 4% paraformaldehyde and permeabilized with 0.3% Triton X-100 (Pierce). Primary antibodies were incubated overnight. Secondary antibodies were Alexa Fluor from Molecular Probes and were incubated for 2 h at room temperature at a 1:1000 dilution. Slides were mounted in VectaShield with DAPI (Vector).

**Patient-derived CRC xenografts and tumoroids**. Fragments of primary human colorectal tumors obtained from MARbiobank with the informed consent of patients and following all recommendations of Hospital del Mar' Ethics Committee, the Spanish regulations, and the Helsinki declaration's Guide were transplanted and expanded in the cecum of nude mice as orthoxenografts. For tumoroids generation, xenografted tumors were disaggregated in 1 mg/mL collagenase II (Sigma) and 20 μg/mL hyaluronidase (Sigma), filtered in 100 μm cell strainer, and seeded in Matrigel (BD Biosciences) as previously described[37].

**Tumoroid viability assays**. A total of 600 single tumoroid cells were plated in 96-well plates in Matrigel. After 4 days in culture, the growing tumoroids were treated with Jag1 or Dll4 blocking antibodies for 72 h at the indicated concentrations. Cell viability was determined using the CellTiter-Glo® 3D Cell Viability Assay (Promega) following manufacturer's instructions in an Orion II multiplate luminometer (Berthold detection systems). All the experiments were conducted in triplicate.

**Retroviral transduction of mouse intestinal tumoroids**. Mouse intestinal tumoroids grown in Matrigel were mechanically disrupted using a 30G needle. Clumps were collected by centrifugation at 900×g for 5 min and cells were further individualized in trypsin for 5 min at 37 °C. Cellular suspension was then centrifuged at 900×g for 5 min, resuspended in 250 μL of retroviral solution containing Polybrene, Y-27632 and either pMIG-ires-GFP control or the pMIEV-Mfng-ires-GFP vector (provided by Dr. Cynthia Guidos) and seeded in 48-well plates. The whole plate was then centrifuged at 600×g at 32 °C for 1 h and incubated for 6 h at 37 °C. Finally, comparable numbers of control or Mfng-transduced cells were plated in Matrigel.

**Human colorectal samples**. Formalin-fixed, paraffin-embedded tissue blocks of colorectal tumors were obtained from Parc de Salut MAR Biobank (MARBiobanc), Barcelona. Multiple areas of invasive carcinoma, adenomas, and normal adjacent mucosa from the same surgical sample were identified on corresponding hematoxylin–eosin-stained slides. Tissue blocks were transferred to a recipient "master" block using a Tissue Microarrayer. A minimum of 2–3 cores from the same tumor was included in the array and each core was 0.6 mm wide spaced,

0.7–0.8 mm apart. A pathologist specialist in the gastrointestinal tract (M.I.) performed the histological characterization of human intestinal specimens.

**IHC and IF**. Intestinal samples or organoid pellets included in the paraffin blocks were sectioned at 4 μm and 2.5 μm, respectively. After deparaffinization and rehydration, endogenous peroxidase activity was quenched (20 min, 1.5% $H_2O_2$) and antigen retrieval was performed depending on the antibody. All primary antibodies were diluted in PBS containing 0.05% BSA and incubated overnight at 4 °C, unless otherwise indicated. Sections were then incubated with specific horse-radish peroxidase-labeled secondary antibody (DAKO Envision), and staining was developed using diaminobenzidine peroxidase substrate kit (DakoCytomation) or Tyramide Signal Amplification System (Perkin Elmer). BrdU (1:250, abcam ab6326) was incubated 2 h at room temperature and the secondary antibody system used was a biotinylated anti-rat antibody (Dako, E0468) 1 h at room temperature followed by the Vectastain ABC kit (Vector, PK6100). Staining was developed as described above. Alcian Blue and H&E staining were performed using standard procedures.

**Image analysis and statistical methods**. IHC of intestinal sections were observed in an Olympus BX61 microscope and blindly assessed as low (0–1+) or high (2–3 +) in the case of JAG1 and Notch, or negative/positive for MFNG by two independent researchers. Images were taken using the cellSens Digital Imaging software. IF images of intestinal sections and the organoids and spheroids were taken by confocal microscopy with a Leica SP5 TCS upright microscope and the Leica Application Suite Advanced Fluorescence software. Kaplan–Meier survival curves were generated using the Prism software and comparisons made using the Log-rank (Mantel–Cox) two-tailed test.

**Analysis of public database**. Normalized expression (RNA-seq RPKMs) data of colorectal tumors in the TCGA cohort and clinical data of patients of the same cohort were downloaded from synapse (syn300013) and TCGA Data Portal (https://tcga-data.nci.nih.gov/tcga/), respectively. These data were used for the analysis of the expression levels of Notch1, MFNG, and known targets of the Notch pathway. To carry out these analyses, we first selected the half of the CRC cohort with higher expression of Notch1, and then separated this subgroup of tumors in two halves according to their expression of MFNG.

**Data availability**. Gene expression data of TCGA colorectal tumors used in this study are available at http://firebrowse.org/?cohort = COADREAD.

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

## Acknowledgements

We thank Jéssica González Miranda, Marta Garrido, and Laura Solé for technical assistance, Cynthia Guidos for providing the expression vector for MFNG and scientific advice, and all the members of the Bigas and Espinosa Laboratories for helpful critical discussions. Antonio Berenguer from the Biostatistics/Bioinformatics Unit of IRB (Barcelona) helped us with the analysis of public databases. The results published here are in part based upon data generated by the TCGA Research Network: http://cancergenome.nih.gov/. This work was supported by grants from Instituto de Salud Carlos III FEDER (PIE15/00008 and PI16/00437), AGAUR 2017 SGR135, and the "Xarxa de Bancs de tumors" sponsored by Pla Director d'Oncologia de Catalunya (XBTC).

## Author contributions

E.L.-A., V.R., C.C., A.V., and B.Y. performed the in vitro experiments and the in vivo genetic deletion of Jag1 experiments. A.S., J.H.C., and M.R.J. performed and analyzed the in vivo treatment with anti-JAG1 antibodies in mice. A.V. performed the in vivo experiments with patient-derived tumors. A.G.-P. analyzed the public genomic data. M.I. analyzed the human tumors and determined the effect of drug treatment. F.T. performed

the statistical analysis of the data. J.A., A.V., A.B., C.W.S., and L.E. designed the project, analyzed the data, supervised the experiments, and wrote the manuscript.

## Additional information

**Competing interests:** A.S., J.H.C., M.R.J., and C.W.S. are or were employed by Genentech Inc., which has commercial interests in some of the molecules described.

