## [Peer Review File · Nature Communications]

Reviewers' comments:

Reviewer #1 Expert in colorectal cancer and cancer stem cells:

Lopez-Arribillaga E. et al. study is aimed to demonstrate the role of Manic Fringe (MFNG) and Jag1 in contributing to the regulation of the Notch signaling pathway in CRC. By using many in vivo models of CRC, (transgenic mouse model, 3D tumor spheroids and patient-derived ortho-xenografts) the authors demonstrated that Jag1 is required for tumor initiation originated from β -catenin-driven adenoma, which promotes stemness and prevents differentiation, while Manic Fringe (MFNG) has a protective role.

Moreover, given that ICN1, Jag1 and MFNG levels predict CRC prognoses, they propose MFGN as a prognostic biomarker in CRC and the antibody anti-Jag1 as a novel therapeutic approach, especially in JAG-high and ICN1-high/MFNG-low, CSM2 CRC subset patients.

Overall, the study is well organized, supported by strong and valid data and it adds novel insights into the molecular regulation of the Notch signaling pathway in CRC in its early stage.

This reviewer would suggest the following minor comments:

Minor comments:

- In Fig. 1G, the authors should describe how many mice were used for each group. The statistical significance is missing.
- Regarding Fig. 2B, the authors should improve the image quality, in particular, of the jag 1 lox/lox staining and they should use a magnification box to better appreciate the results.
- Fig. 3D, what happens when a Tamoxifen treatment is performed for longer than 48 hours? Are the mRNA levels reduced even more? The authors should show the prolonged treatment effect, up to 5 days.
- Fig. 3E, the authors should perform a co-expression of all the markers shown (Ki67, Lyz1, Muc2, CAII) with Active-casp3 because it would be interesting to evaluate the population phenotype that survived.
- Fig. 3, should include an IHC analysis of caspase dependent cell death in jag1lox/lox mouse models.
- In Fig. 6C, it would be useful to include the images of the tumors before the excision.
- In Fig. 6F, it is very important to provide controls of the healthy mucosa (IHC analysis for the morphology and stemness markers in the xenograft tumors). Moreover, following treatment with anti-Jag1, the authors should show data to validate the absence of systemic toxicity.
- The authors should provide more details in the Material and Methods section, specifically about the in vivo experiments (i.e. size of the animal cohort and statistical significance).
- It would be nice to insert more comments about the data regarding caspase-dependent cell death and cell differentiation in Results and Discussion sections.
- It may be also very useful to draw a model (graphical abstract) that explains the ligands-receptor interaction of the Notch signaling pathway in CRC.
- Please carefully check the panel subheadings (A,B,C..)and assure yourself that they match with the figure legends (i.e. in Fig. 6 all the panels subheadings are not correctly matched: panel (A) is not what is described in Figure legends as panel (A) etc..).

Reviewer #2 Expert in cancer stem cells:

The work by Lopez-Arribillaga et al aims at elucidating the role of Jag1-dependent Notch1 receptor regulation in intestinal adenomas driven by mutations in Apc, and the dependency of this regulation on the Notch-modifier Manic Fringe (MFNG). They find that Apc driven tumor development is perturbed if Jag1 is targeted genetically or with antibodies, and demonstrate with in vivo in vitro data, that adenomatous epithelium becomes addicted to Jag1 expression while concurrently loosing expression of MFNG. Whereas it has been previously shown that Notch signaling is required for the maintenance of adenomas (van Es et al. 2005) with a suggested link

to Jag1 ligand binding (Rodilla et al 2009), the here presented effects of Jag1 depletion for tumor development and the correlation of Jag1 and Manic Fringe expression levels is of relevance and the work illuminates a potential therapeutic strategy for a subset of colon cancer patients. However, the manuscript lacks mechanistic insights on the molecular level for the APC dependent reduction in MFNG expression, and on a cellular level for the cell-cell interactions facilitated by the Notch signaling. In addition, several other concerns remain:

Major points:

It is shown that Dll4 and Jag1 localize at the membrane of normal organoids and adenoma derived spheroids (Fig. 4A). These findings suggest that addiction of adenomas for Jag1 to drive Notch-signaling is not caused by reduced availability of Dll-ligands, but by alterations in the specificity of the Notch receptor that favor Jag1 binding. However, the suggested mechanism is not addressed to show that Dll4 cannot induce Notch-signaling in adenomatous epithelium. The paper would also benefit from studies on Dll4/Jag1 dual stainings, and from careful analysis of their expression on the cell type and subcellular level.

The presented role of MFNG for Jag1 dependent Notch-signaling is at this stage correlative. The authors state: "the ability of Notch to respond to Jagged or Delta ligands in the adenomas is mostly defined by MFNG expression, as we have found in the mouse system", which is not demonstrated. To address this, experiments with altered MFNG expression or activity, with subsequent readouts on Dll4 and Jag1 effects, stemness, and spheroid phenotypes are required. In Figure 1 no difference is observed upon treating Apc^{min} mice with Jag1 antibody in the large intestine which is where humans do get colon cancer, the authors should comment about that. Authors show that deleting Jag1 inhibits spheroid formation in vitro and adenoma formation in vivo. They conclude that Jag1 deletion affects Tumor Initiating Cells (TICs). However, authors do not test whether this is a primary effect on the TICs or on tumor niche cells (for example Fatrai et al. 2016) that support TICs growth. This also relates to the general shortcoming of the manuscript to address the cell-cell signaling aspect of Notch-signaling. The whole manuscript does not address the reciprocal localization of Notch1/MFNG and Dll4/Jag1 in stem cells or TICs, and in their neighboring niche cells (Paneths in normal epithelium). Additional experiments also addressing the cellular composition are necessary, as despite the lack of gross alterations, Jag1 deletion on wt epithelium could reduce the number of (Lgr5⁺) intestinal stem cells, and thereby reduce the number of potential TICs.

Minor points:

Figure 2A: A normalized log₂ scale would help to compare the fold reductions

Figure 2D: Why is the differentiated cells marker CA-II down regulated?

Figure 3B: For comparison reasons in spheroid growth a WT minus-Cre mouse sample should be included

Figure 3D: Secretory gene markers should be included as positive controls

Figure 3E: The MUC2/CAII staining is difficult to appreciate and a signal intensity quantification for all markers should be included

Figure 3H: Insufficient data are presented to support the non-toxicity of Jag1 AB. A first approach would be to include a dose dependence curve at Fig. 3H.

Reviewer #3 Expert in Notch signalling:

López-Arribillaga and colleagues address in this study an interesting and poorly explored question in intestinal tumorigenesis: which ligands activate Notch signalling in CRC?

The authors continue and extend their previous observations on the role of the Notch ligand Jagged1, a transcriptional target of Wnt/Bcatenin signalling, as the crucial ligand ensuring Notch activation in intestinal adenomas (Rodilla et al., PNAS 2009). In the present work they refine their

previous analysis using conditional Jagged1 KO mice and conclude that Manic Fringe is responsible for dictating which ligands (Dll or Jagged) activate the Notch pathway. Based on studies in *Drosophila* indicating that the glycosyltransferase Fringe favours the interaction of Notch with the Delta ligand, they assume that the same mechanism underlie the switch from a preferential Notch1-Dll4 interaction in normal intestinal crypts (promoted by Manic Fringe expression) toward a preferential Notch activation by the Jagged1 ligand in intestinal tumours lacking Manic Fringe expression.

Overall, this is a hypothesis-driven analysis that does not definitively demonstrate that Jagged1 deletion or pharmacological inhibition genuinely phenocopy loss of Notch activity, as the results presented do not provide sufficient depth to draw the strong conclusions proposed by the authors. Many points are only superficially addressed and the intriguing possibility of using the expression of Manic Fringe to discriminate between Delta or Jagged-mediated Notch signalling as a prognostic value in human tumours should be further explored and strengthened by addressing the concerns I list here below.

1- When the authors revisit the deletion of Jagged1 in intestinal tumours, they show that intestinal-specific deletion of one Jag1 allele (Jag1lox/+) results in a decrease in the number of tumours from about 25 to 20 (Fig. 1A), whereas the same authors had previously found that ApcMin/+ mice in a Jag1 heterozygous background (not gut-specific) would present a more dramatic reduction of the number of tumours, compared to ApcMin Jag1+/+ mice (decrease from 50 to 20 tumours in their previous PNAS paper). These differences might reflect the importance of stromal Jag1 in tumorigenesis and do not support the sentence "epithelial Jag1 is specifically required for tumor initiation" at page 6.

In view of the observed effect of the loss of a single Jag1 allele on tumour formation (Fig. 1B), it would be more appropriate to analyse the villin-Cre/Jag1lox/+ heterozygous mice along with the homozygous deletion in the subsequent analyses of downstream regulation shown in Fig. 2.

2- Pharmacological Jag1 inhibition shows a beneficial effect only on small tumours (Fig. 1G), which is inconsistent with the trend in better survival shown in Fig. 1H, and the effect of anti-Jag1 antibody treatment is exclusively observed in small intestinal tumours and not in the colon. Do the authors have any explanation for these differences between a genetic deletion and the pharmacological inhibition? Immunostaining anti-MFNG in the normal intestine and in the colon might help explaining these discrepancies. Is MFNG highly expressed in the colon, favouring interaction between Notch1 and Dll4? Also, does anti-Jag1 treatment reduce the levels of active Notch1 (ICN1), as shown in Fig. 1C for the genetic Jag1 deletion?

3- The analysis of tumour spheroids in vitro is then conducted with a different mouse line, Bactin-CreERT2. The authors find that in vitro Jag1 loss leads to a rapid organoid death (within 48h in Fig. 4D or 72 hr in Fig. 3C upon Cre induction), although surprisingly 48h upon KO induction in Fig. 3E, the organoids present a relatively normal morphology, with still some dividing cells (Ki67+). The effect of Notch inhibition with DAPT (shown in Fig. S2A) should be shown in Fig. 3C alongside Jag1 deletion, for comparison; the IF staining should also be performed in parallel, to determine if indeed the effect of Jag1 deletion recapitulates Notch pathway inhibition.

4- The IF experiments are often not convincing. ICN1 staining should be shown in the normal intestine, to appreciate the expected crypt specific expression of nuclear Notch. Fig. S1F does not show any detectable Notch2 nuclear staining and the difference between WT and KO animals is not clear. Fig. S1D is out of focus. The images presented in Fig. S2C on "normal organoids" show no dividing cells (Ki67+) in the crypt regions of the organoids and the MUC2 and CAII staining do not localise as expected either.

5- The morphology of "normal organoids" in Fig. S2C looks aberrant; this may be due to the fact that the authors culture both organoids and tumour spheroids in the presence of Wnt3A and R-

spondin. Wnt3A is not required for the growth of small intestinal organoids and R-spondin should not be used in tumour cultures, as Apc mutant tumour spheroids are reported to grow independently of R-spondin.

6- The same remark about non conclusive IF experiments applies to the immunostaining analyses presented in Fig. 4A. At the low magnification presented, it appears that the same cells in both normal organoids and spheroids co-express Jag1 and Dll4. Dll4 expression should be restricted to Paneth cells in normal organoids, but higher magnifications are required to appreciate the differences in expression in different cell types. Moreover, a double immunofluorescence anti-ICN1/Jag1 and anti-ICN1/Dll4 would be important to define which cells express the ligands and which cells the receptors, as a well-established negative feedback mechanism should prevent co-expression of receptor and ligand in the same cell.

These experiments should also be corroborated by immunostaining of both normal intestine and tumours for Jag1 and Dll4, possibly in co-staining analysis with ICN1 or in serial sections.

7- In the WB of Fig. 4C I don't understand why WT Jag1 is still present, along with the exon4-deleted form, upon 4-OHT treatment. How long after 4-OHT administration was the experiment performed? If it is 72h as shown in the bright field images in the same Figure, it is not surprising if the authors do not see a phenotype, as WT Jag1 was still present.

8- The analysis of human carcinomas is difficult to interpret for the following reasons:

a) If Jag1 is indeed a target of Wnt/Bcatenin activation, why some carcinomas express no Jag1 (as shown in Fig. 5A)? Immunostaining for Jag1 and Bcatenin might help clarify this point.

b) In carcinomas showing no Jag1 expression and MFNG+, is Dll4 expressed? This should be tested in order to support the conclusion that, in these tumours, Jag1 inhibition would not be beneficial, as Notch1 would be activated by Dll4. Is there no Dll4+ expression in CRC tumours or do the authors assume a different signaling output if Notch1 is activated by Dll4 instead of Jag1?

c) More importantly, the authors claim an inverse correlation between Jag1 and ICN1 in MFNG+ tumours; why the presence of MFNG in certain tumours should influence Jag1 expression? If we follow the authors' hypothesis, MFNG should induce Notch1 glycosylation, hence favouring Notch1 interaction with Dll4, but this should not affect Jag1 expression.

d) A worst prognosis is predicted when tumours are Jag1Hi and MFNGneg (Fig. 5C); how do ICN1 levels correlate with disease free survival? Jag1Hi might simply reflect strong Wnt/Bcatenin activation, regardless of Notch activation, echoing my main concern about the entire work, which may be looking at a Jag1-mediated effect not specific to Notch signalling.

e) Fig. 5E is unclear: "Levels of MFNG in this subgroup of tumors also determined the transcriptional status of several Notch target genes known to be relevant for cancer progression". What is the difference between Up and Down genes and how does it correlate with Notch target genes and with MFNG Hi or Low tumours? Are these real Notch target genes? In which context?

f) The analysis of human tumour samples shows that MFNG is more often expressed in carcinomas than in adenomas (198 vs. 41 in Fig. S4B), whereas ICN1 is more often expressed in adenomas (92 ICN1Hi vs. 43 ICN1Low in Fig. S2A) than in carcinomas (106 ICN1Hi vs. 136 ICN1Low). Jag1 expression does not seem to significantly cluster towards a specific tumour type, so the insights from this analysis are unclear and percentages of tumours should be presented to clarify the point the authors want to make.

9- The effect of anti-Jag1 treatment on tumor xenografts presented in Fig. 6 is in apparent contradiction with the results obtained in mouse endogenous tumours, where Jag1 inhibition showed no effect in big tumours (> 2.5mm) but only on small polyps (Fig. 1G). If this were conserved in human tumours, no beneficial effect of anti-Jag1 antibody treatment should be observed in human carcinomas that were treated 20 days after caecum transplant, therefore not comparable to the small nascent adenomas of Fig. 1G.

A critical experiment that could nicely validate the authors' hypothesis would imply to implant a human tumour selected among the MFNG Hi tumours, as in this context the anti-Jag1 antibody

should show no effect on tumour growth, since Dll4/Notch signaling would be at work.

Minor corrections:

- the legend to Fig. 6 is incorrect. 6A=6B, 6B=6C, 6C=6D, 6D=6F, 6E=6F and 6F=6E. As there is no legend to 6A, it is unclear what the red staining represents in this panel.
- comparable tumours should be shown in Fig. 2D.
- RT-PCR for the well-established direct Notch target gene Nrarp should be shown in Fig. 1D, as both c-Myc and Hes1 can be activated by Wnt signaling as well.
- there is no indication of the antibody used to detect Notch2 in Fig. S1F.
- the images in Fig. 6F show an increased fibrosis or inflammation in the presence of the anti-Jag1 Ab, but not necessarily increased apoptosis. The decrease in Ki67+ cells is also not obvious in the images presented in the upper panel.
- the relevant literature in the Introduction is not comprehensively cited.

Reviewers' comments:

Reviewer #1 Expert in colorectal cancer and cancer stem cells:

Lopez-Arribillaga E. et al. study is aimed to demonstrate the role of Manic Fringe (MFNG) and Jag1 in contributing to the regulation of the Notch signaling pathway in CRC. By using many in vivo models of CRC, (transgenic mouse model, 3D tumor spheroids and patient-derived ortho-xenografts) the authors demonstrated that Jag1 is required for tumor initiation originated from β -catenin-driven adenoma, which promotes stemness and prevents differentiation, while Manic Fringe (MFNG) has a protective role.

Moreover, given that ICN1, Jag1 and MFNG levels predict CRC prognoses, they propose MFGN as a prognostic biomarker in CRC and the antibody anti-Jag1 as a novel therapeutic approach, especially in JAG-high and ICN1-high/MFNG-low, CSM2 CRC subset patients.

Overall, the study is well organized, supported by strong and valid data and it adds novel insights into the molecular regulation of the Notch signaling pathway in CRC in its early stage.

Answer: We thank the reviewer for these positive comments.

This reviewer would suggest the following minor comments:

Minor comments:

-In Fig. 1G, the authors should describe how many mice were used for each group. The statistical significance is missing.

Answer: We apologize for omitting this data that is now included in the text and in the figure.

- Regarding Fig. 2B, the authors should improve the image quality, in particular, of the jag 1 lox/lox staining and they should use a magnification box to better appreciate the results.

Answer: We have improved the figure with new and better resolution in situ images.

Fig. 3D, what happens when a Tamoxifen treatment is performed for longer than 48 hours? Are the mRNA levels reduced even more? The authors should show the prolonged treatment effect, up to 5 days.

Answer: This analysis cannot be done since tumoroids start dying after 72 hours of treatment and there are no cells alive at 5 days. We have chosen 48 hours since it is the time point in which cells are still in good conditions.

Fig. 3E, the authors should perform a co-expression of all the markers shown (Ki67, Lyz1, Muc2, CAII) with Active-casp3 because it would be interesting to evaluate the population phenotype that survived.

Answer: We have performed new staining for figure 3E using tumoroid sections. With the resolution that we have now obtained it is clear that apoptotic cells accumulate in the lumen of the sphere colocalizing with Alcian Blue-positive cells, most likely mucus-secreting Goblet Cells. In the case of ki67, we never detected ki67 staining in the lumen of the spheres where caspase 3 is found. Interestingly, we noticed a strong parallelism between OH-TAM treatment and DAPT treatment as it is now shown in new Figure 3E.

Fig. 3, should include an IHC analysis of caspase dependent cell death in jag1lox/lox mouse models.

Answer: We performed cleaved-caspase 3 staining in the Jag1lox/lox mice at the time of sacrifice and we did not detect any difference between the wildtype and knockout animals, suggesting that Jag1 deletion acts in the first stages of tumor formation and, in addition, does not affect normal intestinal cells.

- In Fig. 6C, it would be useful to include the images of the tumors before the excision.

Answer: We have included these images in new Figure S5.

- In Fig. 6F, it is very important to provide controls of the healthy mucosa (IHC analysis for the morphology and stemness markers in the xenograft tumors). Moreover, following treatment with anti-Jag1, the authors should show data to validate the absence of systemic toxicity.

Answer: We are now including staining for Ki67 and active-caspase 3 of the

normal mouse intestinal tissue adjacent to the xenograft from both control and anti-JAG1 treated animals in new Figure 6G.

- The authors should provide more details in the Material and Methods section, specifically about the in vivo experiments (i.e. size of the animal cohort and statistical significance).

Answer: We have expanded this information in the Results and Methods sections.

- It would be nice to insert more comments about the data regarding caspase-dependent cell death and cell differentiation in Results and Discussion sections.

Answer: We have included a comment about the possible relation between differentiation and apoptosis in Jag1 deleted cells in page 8: "... active-Caspase 3 and Alcian blue staining was colocalizing in cells delivered into the spheroid lumen suggesting that Notch inactivation following Jag1 deletion imposes the terminal mucosecretory differentiation of adenoma cells leading to cellular apoptosis."

- It may be also very useful to draw a model (graphical abstract) that explains the ligands-receptor interaction of the Notch signaling pathway in CRC.

Answer: We thank the reviewer for this suggestion. We have included a model in new Figure 7.

- Please carefully check the panel subheadings (A, B, C..) and assure yourself that they match with the figure legends (i.e. in Fig. 6 all the panels subheadings are not correctly matched: panel (A) is not what is described in Figure legends as panel (A) etc..).

Answer: We thank the reviewer for the indication. We have checked it and corrected.

Reviewer #2 Expert in cancer stem cells:

The work by Lopez-Arribillaga et al aims at elucidating the role of Jag1-dependent Notch1 receptor regulation in intestinal adenomas driven by mutations in Apc, and the dependency of this regulation on the Notch-modifier Manic Fringe (MFNG). They find that Apc driven tumor development is perturbed if Jag1 is targeted genetically or with antibodies, and demonstrate with in vivo in vitro data, that adenomatous epithelium becomes addicted to Jag1 expression while concurrently loosing expression of MFNG. Whereas it has been previously shown that Notch signaling is required for the maintenance of adenomas (van Es et al. 2005) with a suggested link to Jag1 ligand binding (Rodilla et al 2009), the here presented effects of Jag1 depletion for tumor development and the correlation of Jag1 and Manic Fringe expression levels is of relevance and the work illuminates a potential therapeutic strategy for a subset of colon cancer patients.

However, the manuscript lacks mechanistic insights on the molecular level for the APC dependent reduction in MFNG expression, and on a cellular level for the cell-cell interactions facilitated by the Notch signaling.

Answer: As the reviewer noticed, the main goal of this work is to understand the Notch dependency of some tumors but not others. We found that the different usage of Jag1 or Dll4 because of MFNG expression is playing a major role in this effect. We do not have any evidence that APC mutation imposes a reduction in MFNG levels, in fact most tumors have active Wnt pathway and only few have reduced levels of MFNG. In any case, we believe that this investigation is out of the scope of this work.

Other concerns:**Major points:**

It is shown that Dll4 and Jag1 localize at the membrane of normal organoids and adenoma derived spheroids (Fig. 4A). These findings suggest that addiction of adenomas for Jag1 to drive Notch-signaling is not caused by reduced availability of Dll-ligands, but by alterations in the specificity of the

Notch receptor that favor Jag1 binding. **However, the suggested mechanism is not addressed to show that Dll4 cannot induce Notch-signaling in adenomatous epithelium.**

Answer: To check whether Dll4 can induce Notch1 signaling in the adenoma cells, we performed double IF with ICN1 (active Notch) and Dll4 in the JAG1-deleted tumoroids (that express Dll4). As it is now shown in new Figure S2D OH-TAM treated spheroids still express Dll4 but lack ICN1. We also confirm that this adenoma cells show reduced levels of the Notch-targets Hes1 and Nrarp (new figure 3D).

-The paper would benefit from studies on Dll4/Jag1 dual staining and from careful analysis of their expression on the cell type and subcellular level.

Answer: We confirmed that both Jag1 and Dll4 are expressed along the monolayer of these structures. Based on the distribution of Jag1 and Dll4 in the organoids/spheroids we do not think that including double staining will contribute to improve the manuscript. The same is observed in human tumors that express both ligands. These experiments are included in new Figures 4A and S2D.

-The presented role of MFNG for Jag1 dependent Notch-signaling is at this stage correlative. The authors state: “the ability of Notch to respond to Jagged or Delta ligands in the adenomas is mostly defined by MFNG expression, as we have found in the mouse system”, which is not demonstrated. To address this, **experiments with altered MFNG expression or activity, with subsequent readouts on Dll4 and Jag1 effects, stemness, and spheroid phenotypes are required.**

Answer: We agree with the reviewer that this is a crucial experiment, although final number of cells is scarce for exhaustive examination. We attempted to do the proposed characterization with the organoids obtained from the Jag1-deletion rescue experiments that we did with exogenous MFNG, however, the amount of transduced cells (even after cell sorting) has been insufficient to perform the suggested characterization. Of note, we could not see any morphological difference between MFNG positive and negative spheroids before Jag1 deletion or with rescued organoids. This observation that is now

included in page 9 suggests that the main effect of MFNG is to provide the ability to activate Notch in the absence of Jag1, but not the specificity of the Notch response. However, we sincerely think that elucidating the Notch-dependent functional differences that arise from the differential usage of Jag and Dll ligands in the intestinal cells is extremely interesting but out of the scope of this work. In this sense, preliminary analysis of CRC cell lines with variable levels of MFNG, Jag1 and Dll4 do not show a clear correlation between Jag1 or Dll4 usage with transcriptional activity of Notch-dependent genes but also suggest that the best combination to induce intracellular Notch1 is low-MFNG and high-Jag1 (see Figure below).

In Figure 1 no difference is observed upon treating ApcMin mice with Jag1 antibody in the large intestine, which is where humans do get colon cancer, the authors should comment about that.

Answer: We have carefully revised our data and we should emphasize that tumors are practically lost in the large intestine of the anti-Jag1 treated animals. However, as it is shown in the figure 1F, differences between controls and anti-Jag1 treated mice are not significant because of the low number of colonic tumors that naturally occur in the APC^{Min} background. We have now included this observation in the text for clarity. In page 6: "After ten weeks of weekly treatment, selective Jag1 blockade significantly reduced the number of polyps in the small intestine (SI) and lowered the number of colonic (CO) tumors from 1.7 to 0.45 per animal in average (Figure 1F), with no detectable signs of systemic toxicity. However, effect of anti-Jag1 in colonic tumorigenesis did not reach statistical significance likely due to the basal low incidence of these

tumors in the Apc^{Min/+} background.”

R: Authors show that deleting Jag1 inhibits spheroid formation in vitro and adenoma formation in vivo. They conclude that Jag1 deletion affects Tumor Initiating Cells (TICs). However, authors do not test whether this is a primary effect on the TICs or on tumor niche cells (for example Fatrai et al. 2016) that support TICs growth. This also relates to the general shortcoming of the manuscript to address the cell-cell signaling aspect of Notch-signaling. The whole manuscript does not address the reciprocal localization of Notch1/MFNG and Dll4/Jag1 in stem cells or TICs, and in their neighboring niche cells (Paneths in normal epithelium). **Additional experiments also addressing the cellular composition are necessary, as despite the lack of gross alterations, Jag1 deletion on wt epithelium could reduce the number of (Lgr5+) intestinal stem cells, and thereby reduce the number of potential TICs.**

Answer: We have now tested two different commercially available antibodies against Lgr5+ (from Abcam) to determine whether this particular stem cell population is altered in the adenoma tissue and in the non-transformed crypts of the JAG1 KO animals. These experiments are now included in new Figure 2D and demonstrate that both antibodies detect a similar number of Lgr5+ cells in the non-transformed intestinal crypts whereas the number of positive cells is clearly reduced in the adenoma tissue of Jag1 KO mice compared to WT adenomas. In the text in page 7: “IHC analysis of LGR5 protein with a specific antibody confirmed these results, and excluded a defect in the normal intestinal stem cell compartment of Jag1 KO mice (Figure 2D).”

Minor points:

Figure 2A: A normalized log2 scale would help to compare the fold reductions

Answer: We have tried changing the scale but we do not see any improvement. So we prefer maintaining the original format that is the same for all other figures.

Figure 2D: Why is the differentiated cells marker CA-II down regulated?

Answer: We do not have an answer but we found the same effect in the spheroid model in Figure 3E. Our hypothesis is that adenoma cells tend to differentiate into the secretory lineages (Paneth or Goblet cells) in the absence of Jag1.

Figure 3B: For comparison reasons in spheroid growth a WT minus-Cre mouse sample should be included.

Answer: We always try to compare different genotypes all containing the Cre to exclude phenotypes that are mediated by the recombinase, which have been previously reported. However, we have not detected any difference when growing controls or Cre-expressing adenoma cells, such is the case of the inducible Jag1^{+/+} spheroids shown in 3C.

Figure 3D: Secretory gene markers should be included as positive controls.

Answer: These controls were already shown in Figure 3E.

Figure 3E: The MUC2/CAII staining is difficult to appreciate and signal intensity quantification for all markers should be included

Answer: We have changed the muc2 staining for the more general Alcian Blue staining. CAII is difficult to detect but we have improved the images to show that it is not increased.

Insufficient data are presented to support the non-toxicity of Jag1 AB. A first approach would be to include a dose dependence curve at Fig. 3H.

Answer: We have included this data in 3H as suggested. In addition, we are showing the cleaved-caspase3 and ki67 staining from anti-JAG1 treated mice in Figure 6G.

Reviewer #3 Expert in Notch signaling:

López-Arribillaga and colleagues address in this study an interesting and poorly explored question in intestinal tumorigenesis: which ligands activate Notch signaling in CRC?

The authors continue and extend their previous observations on the role of the Notch ligand Jagged1, a transcriptional target of Wnt/ β -catenin signaling, as the crucial ligand ensuring Notch activation in intestinal adenomas (Rodilla et al., PNAS 2009). In the present work they refine their previous analysis using conditional Jagged1 KO mice and conclude that Manic Fringe is responsible for dictating which ligands (DII or Jagged) activate the Notch pathway. Based on studies in *Drosophila* indicating that the glycosyltransferase Fringe favors the interaction of Notch with the Delta ligand, they assume that the same mechanism underlie the switch from a preferential Notch1-DII4 interaction in normal intestinal crypts (promoted by Manic Fringe expression) toward a preferential Notch activation by the Jagged1 ligand in intestinal tumors lacking Manic Fringe expression.

Overall, this is a hypothesis-driven analysis that does not definitively demonstrate that Jagged1 deletion or pharmacological inhibition genuinely phenocopy loss of Notch activity, as the results presented do not provide sufficient depth to draw the strong conclusions proposed by the authors. Many points are only superficially addressed and the intriguing possibility of using the expression of Manic Fringe to discriminate between Delta or Jagged-mediated Notch signaling as a prognostic value in human tumors should be further explored and strengthened by addressing the concerns I list here below.

Answer: We thank the reviewer for the constructive comments and will address all the points to convince that Manic fringe discriminates between Delta and Jag-mediated Notch signaling in human colon tumors.

1- When the authors revisit the deletion of Jagged1 in intestinal tumors, they show that intestinal-specific deletion of one Jag1 allele (Jag1lox/+)

results in a decrease in the number of tumors from about 25 to 20 (Fig. 1A), whereas the same authors had previously found that ApcMin/+ mice in a Jag1 heterozygous background (not gut-specific) would present a more dramatic reduction of the number of tumors, compared to ApcMin Jag1+/+ mice (decrease from 50 to 20 tumors in their previous PNAS paper). These differences might reflect the importance of stromal Jag1 in tumorigenesis and do not support the sentence “epithelial Jag1 is specifically required for tumor initiation” at page 6.

Answer: We totally agree with the reviewer that stromal JAG1 may also contribute to intestinal tumorigenesis, which is not surprising. Thus, we have modified the above-mentioned sentence now in page 5 to: “These results are different to that obtained using the general Jag1 heterozygous mice and indicate that epithelial Jag1 is specifically required for tumor initiation. However, Jag1 expressed in the stroma of the tumor may also contribute to different aspects of tumor initiation and progression, as suggested from our previous work²⁹.”

In view of the observed effect of the loss of a single Jag1 allele on tumor formation (Fig. 1B), it would be more appropriate to analyze the villin-Cre/Jag1flox/+ heterozygous mice along with the homozygous deletion in the subsequent analyses of downstream regulation shown in Fig. 2.

Answer: We have included several Jag1 heterozygous mice in most of the experiments and we observed that the phenotypes were similar to the WT (Fig. 3B) or intermediate (in between WT and KO), and we already included some data when relevant (i.e. 1B, 2D). However, we think that including heterozygous in all the experiments could provide interesting information about Jag1 dosage, however it would not contribute to the clarity of the message.

2- Pharmacological Jag1 inhibition shows a beneficial effect only on small tumors (Fig. 1G), which is inconsistent with the trend in better survival shown in Fig. 1H, and the effect of anti-Jag1 antibody treatment is exclusively observed in small intestinal tumors and not in the colon. Do the authors have any explanation for these differences between a genetic deletion and the pharmacological inhibition?

Immunostaining anti-MFNG in the normal intestine and in the colon might help explaining these discrepancies. Is MFNG highly expressed in the colon, favoring interaction between Notch1 and Dll4? Also, does anti-Jag1 treatment reduce the levels of active Notch1 (ICN1), as shown in Fig. 1C for the genetic Jag1 deletion?

Answer: Genetic and pharmacologic JAG1 interventions are primarily different since accessibility of the “pharmacologic” treatment once tumors are formed is uncertain (likely less accessible in case of big tumors). However, we totally disagree with the supposedly lack of response in the colon, since Figure 1F shows an important reduction in the number of colonic tumors in anti-Jag1 treated animals although it does not reach statistical significance. We have reanalyzed the data and found that lack of significance is likely due to the low number of tumors and the high variability in number that are found in the control animals. Thus we have now rewrite the text to include this observation, in page 6: “After ten weeks of weekly treatment, selective Jag1 blockade significantly reduced the number of polyps in the small intestine (SI) and lowered the number of colonic (CO) tumors from 1.7 to 0.45 per animal in average (Figure 1F), with no detectable signs of systemic toxicity. However, effect of anti-Jag1 in colonic tumorigenesis did not reach statistical significance likely due to the basal low incidence of these tumors in the $Apc^{Min/+}$ background.”

3- The analysis of tumor spheroids in vitro is then conducted with a different mouse line, Bactin-CreERT2. The authors find that in vitro Jag1 loss leads to a rapid organoid death (within 48h in Fig. 4D or 72 hr in Fig. 3C upon Cre induction), although surprisingly 48h upon KO induction in Fig. 3E, the organoids present a relatively normal morphology, with still some dividing cells (Ki67+).

Answer: We thank the reviewer for the indication, and we apologize for this mistake since images shown in Figure 4D were captured after 72h of 4-OH-T treatment. In general, $Jag1^{lox/lox}$ tumoroids treated with 4-OH-T remains apparently healthy 24-48 hours (at the time we performed all transcriptional and IHC analyses), and then start collapsing around day 3-4. We have corrected the label in Figure 4D.

-The effect of Notch inhibition with DAPT (shown in Fig. S2A) should be shown in Fig. 3C alongside Jag1 deletion, for comparison; the IF staining should also be performed in parallel, to determine if indeed the effect of Jag1 deletion recapitulates Notch pathway inhibition.

Answer: We have not considered appropriate to move S2A to 3C but instead we have performed the whole DAPT analysis as suggested, and included the new data in Figure 3E. In fact we have remade the whole figure to obtain better images from paraffin sections. In the text in page 8, we now mention these new results: "By IHC analysis we found that 4-OH tamoxifen treatment considerably diminished the number of proliferating Ki67-positive cells and increased the quantity of lysozyme- (Paneth) and Alcian Blue-positive (mucosecretory) cells without affecting CAII levels (Figures 3E, 3F and 3G). These changes correlated with an increase in the number of apoptotic cells, as determined by active-Caspase 3 staining at 48 hours after 4-OH tamoxifen treatment closely resembling the effects of Notch inhibition by DAPT (Figure 3E)."

4- The IF experiments are often not convincing. ICN1 staining should be shown in the normal intestine, to appreciate the expected crypt specific expression of nuclear Notch.

Answer: We have now repeated most of the non-convincing IF experiments. In the case of ICN1, we are doing this staining routinely and published the suggested control in previous articles such as Rodilla et al, PNAS 2009. However, if the reviewer still thinks that adding this control is essential to support our conclusions we can include these images where required.

Fig. S1F does not show any detectable Notch2 nuclear staining and the difference between WT and KO animals is not clear. Fig. S1D is out of focus. The images presented in Fig. S2C on "normal organoids" show no dividing cells (Ki67+) in the crypt regions of the organoids and the MUC2 and CAII staining do not localize as expected either.

Answer: We apologize for the low quality of the images that we presented in the previous version. We have now repeated these experiments with a new anti-N2 antibody (from Abcam) and included high quality images in new figure S1F.

5- The morphology of “normal organoids” in Fig. S2C looks aberrant; this may be due to the fact that the authors culture both organoids and tumour spheroids in the presence of Wnt3A and R-spondin. Wnt3A is not required for the growth of small intestinal organoids and R-spondin should not be used in tumor cultures, as Apc mutant tumor spheroids are reported to grow independently of R-spondin.

Answer: We apologize for the generalization that we made in the Methods section about organoid and spheroid/tumoroid growth conditions. In fact, our cultures are all based on standard protocols and we only add Wnt3 at the beginning of the organoids cultures to facilitate their formation, and then removed. However, we add R-spondin in the culture media indistinctly for simplicity and to simultaneously test normal and tumoral growth (although we now it is not necessary for most tumoroids cultures). We have now corrected this mistake in the Methods section in page 14: “According to standard protocols, Wnt3 was only added at the start of the organoids cultures to facilitate their formation, and then removed.”

Related with the observation that organoids shown in Figure S2C are aberrant, we have proven expertise in organoid formation, and we always detect multiple morphologies in the cultures, although they tend to form ‘crypt-like structures’ in contrast with cancer cells that form spherical structures.

6- The same remark about non-conclusive IF experiments applies to the immunostaining analyses presented in Fig. 4A. At the low magnification presented, it appears that the same cells in both normal organoids and spheroids co-express Jag1 and Dll4. Dll4 expression should be restricted to Paneth cells in normal organoids, but higher magnifications are required to appreciate the differences in expression in different cell types. Moreover, a double immunofluorescence anti-ICN1/Jag1 and anti-ICN1/Dll4 would be important to define which cells express the ligands and which cells the receptors, as a well-established negative feedback mechanism should prevent co-expression of receptor and ligand in the same cell. These experiments should also be corroborated by immunostaining of both normal intestine and tumors for Jag1 and Dll4, possibly in co-staining analysis with ICN1 or in serial sections.

Answer:

We have now repeated the stainings shown in Figure 4A to improve their quality. We are now including IHC analysis (fluorescent dye) of Jag1 and Dll4 in organoids and spheroids. Again, we detected both ligands all along the epithelial monolayer of these 3D structures what can be surprising but reproducible. Thus, we do not think that including a double staining with lysozyme will be informative.

On the other hand, we did not find any reports on the expression (at the protein levels) of Jag1 and Dll4 in organoids or spheroids, so we do not know why the reviewer says that it should be restricted in Paneth cells. In normal intestine, the reported immunofluorescence analysis does not show that pattern either (Watanabe et al. Peer J. 2014). The reviewer probably refers to the mRNA expression data of Dll4 in Paneth cells compared to Lgr5+ cells (Sato et al, Nature 2011), which does not mean the Dll4 protein is only in Paneth cells.

7- In the WB of Fig. 4C I don't understand why WT Jag1 is still present, along with the exon4-deleted form, upon 4-OHT treatment. How long after 4-OHT administrations was the experiment performed? If it is 72h as shown in the bright field images in the same Figure, it is not surprising if the authors do not see a phenotype, as WT Jag1 was still present.

Answer: We agree that deletion of WT (functional) JAG1 is difficult to see because it relies in the single removal of exon4 involving the Notch binding domain. We have now repeated the experiment but we could not significantly improve the quality of the image due to the low amount of protein that we obtain from the organoid cultures (in particular after 48-72 hours of treatment). Thus we decided to try detecting the deletion by PCR using a pair of primers that specifically amplify the non-deleted band (binding on exon 4). PCR analysis is now included in new figure 4C to avoid further miss-understandings. Nevertheless and to further convince the reviewer we here include a detail of the WB image indicating the location of a non-specific band (&) that appears in the all lanes after long exposure times, instead the Jag1 WT band () is reduced at low 4-OH-T and lost at high 4-OH-T concentrations (72 hours) and the $\Delta E4$ -Jag1 protein (#) that is only detected in 4-OH-T treated samples.*

8- The analysis of human carcinomas is difficult to interpret for the following reasons:

a) If Jag1 is indeed a target of Wnt/ β -catenin activation, why some carcinomas express no Jag1 (as shown in Fig. 5A)? Immunostaining for Jag1 and β -catenin might help clarify this point.

Answer: We have performed this IHC and detected variable levels and percentages of nuclear β -catenin in most of the tumors analyzed with no apparent correlation between nuclear β -catenin and Jag1 in the carcinoma samples. These results suggest that although β -catenin activity is required for Jag1 transcription, as we previously found, it is likely to be not sufficient to sustain high Jag1 protein levels.

b) In carcinomas showing no Jag1 expression and MFNG+, is Dll4 expressed? This should be tested in order to support the conclusion that, in these tumors, Jag1 inhibition would not be beneficial, as Notch1 would be activated by Dll4. Is there no Dll4+ expression in CRC tumors or do the authors assume a different signaling output if Notch1 is activated by Dll4 instead of Jag1?

Answer: We have performed Dll4 staining in a subset of the CRC samples and found that 10 out of 11 ICN1-high/JAG1-negative tumors are Dll4 high, being 9 of them MFNG positive. These results are now included in Supplementary table S2 and mentioned in the text in page 10: "...ICN1 detection was biased towards the JAG1 high group in the absence of MFNG (9 out of 14 MFNG negative and ICN1 high tumors were JAG1 high) (Figure 5B). Importantly, further analysis of a subset of CRC samples showed that 10 out of 11 ICN1 high/JAG1 low tumors contained high levels of Dll4, being 9 of them MFNG positive (Supplementary table S2), supporting the notion that tumors expressing MFNG may induce Notch1 through Dll4."

c) More importantly, the authors claim an inverse correlation between Jag1 and ICN1 in MFNG+ tumors; why the presence of MFNG in certain tumors should influence Jag1 expression? If we follow the authors' hypothesis, MFNG should induce Notch1 glycosylation, hence favoring Notch1 interaction with Dll4, but this should not affect Jag1 expression.

Answer: We apologize for the miss-understanding. We did not deliberately suggested that MFNG was affecting JAG1 expression, and the reasoning followed by the reviewer is totally correct: we find less active Notch (ICN1) in the Jag1high MFNG+ tumors.

However a causal correlation between levels of MFNG and Jag1 cannot be excluded, and even expected from data in Drosophila (Wnt induces Iroquois activity leading to Fringe repression (Dominguez and deCelis, Nature 1998; Maurel-Zaffran and treisman Dev. 2000)).

d) A worst prognosis is predicted when tumours are Jag1Hi and MFNGneg (Fig. 5C); how do ICN1 levels correlate with disease free survival? Jag1Hi might simply reflect strong Wnt/ β -catenin activation, regardless of Notch activation, echoing my main concern about the entire work, which may be looking at a Jag1-mediated effect not specific to Notch signaling.

Answer: This is a very good point and we thank the reviewer for mentioning. To further clarify this issue, we have performed the same patient survival analysis inside the ICN1 negative group. Our results that are shown in new Figure S4C indicate that Jag1 and MFNG have no predictive value in the absence of activated Notch1. We have included this information in the text in page 10: "In contrast, we did not observe any association between JAG1 and/or MFNG levels and patient outcome in the Notch negative population (Figure S4C)."

e) Fig. 5E is unclear: "Levels of MFNG in this subgroup of tumors also determined the transcriptional status of several Notch target genes known to be relevant for cancer progression". What is the difference between Up and Down genes and how does it correlate with Notch target genes and with MFNG Hi or Low tumors? Are these real Notch target genes? In which context?

Answer: We thank the reviewer for the indication and we apologize for the lack

of clarity. Related to the criteria that we followed for selecting these particular Notch targets, they were selected based on published data (references are included) and their differential expression in our groups of interest (MFNG high and low). However, we agree that the figure is somehow confusing, we have changed the figure legend to: “(E) Genes previously identified as Notch targets in the indicated publications that were up- (Up) or down-regulated (Down) in the MFNG-low tumors compared to the MFNG-high.”

f) The analysis of human tumor samples shows that MFNG is more often expressed in carcinomas than in adenomas (198 vs. 41 in Fig. S4B), whereas ICN1 is more often expressed in adenomas (92 ICN1Hi vs. 43 ICN1Low in Fig. S2A) than in carcinomas (106 ICN1Hi vs. 136 ICN1Low). Jag1 expression does not seem to significantly cluster towards a specific tumor type, so the insights from this analysis are unclear and percentages of tumors should be presented to clarify the point the authors want to make.

Answer: The reviewer is correct that ICN1, Jag1 and MFNG expression by itself do not correlate with any of the parameters analyzed and there is no correlation between them, and the reviewer is also correct in the sense that the text is a little confusing. We have now modified the text to make it clear and we also added some data regarding Dll4 expression in a subset of CRC samples.

In page 10: “... ICN1 detection was biased towards the JAG1 high group in the absence of MFNG (9 out of 14 MFNG negative and ICN1 high tumors were JAG1 high) (Figure 5B). Importantly, further analysis of a subset of CRC samples showed that 10 out of 11 ICN1 high/JAG1 low tumors contained high levels of Dll4, being 9 of them MFNG positive (Supplementary table S2), supporting the notion that tumors expressing MFNG may induce Notch1 through Dll4.”

We hope that now the text better reflects that there is no univocal association between Notch1 activation and particular combinations of MFNG and Jag1/Dll4, and that only the combination of ICN1+, JAG1 high and MFNG negative defines a subgroup of poor prognosis patients that we propose as candidates for anti-JAG1 therapy. However, in the case that this information is still confusing, and since we are adding much more information to the final version of the

manuscript, we could move this particular information to the supplementary section.

9- The effect of anti-Jag1 treatment on tumor xenografts presented in Fig. 6 is in apparent contradiction with the results obtained in mouse endogenous tumors, where Jag1 inhibition showed no effect in big tumors (> 2.5mm) but only on small polyps (Fig. 1G). If this were conserved in human tumors, no beneficial effect of anti-Jag1 antibody treatment should be observed in human carcinomas that were treated 20 days after caecum transplant, therefore not comparable to the small nascent adenomas of Fig. 1G.

Answer: We agree with the reviewer and our initial expectation was to observe minor effects on the primary tumors but a decrease in their metastatic capacity. However, the reality was that anti-JAG1 treatment significantly reduced the primary tumors, and since the tumor we assayed failed to produce evident intraperitoneal metastasis in the majority of animals transplanted the results from this part were not statistically valuable.

On the other hand, we have now analyzed the levels of MFNG in the remaining intestinal tumors of the APC^{Min};Jag1 KO mice and detected the presence of MFNG that would impose a superior resistance to anti-Jag1 treatment or deletion. We can speculate that this mechanism of resistance is not working in the human tumor xenograft model? We have included this information in new Figure S3B and in the text in page 9: “MFNG levels were also increased in tumors arising in the Apc^{Min/+} Jag1 KO intestines (Figure S3B) further suggesting that MFNG levels regulate addiction to Jag1.”

A critical experiment that could nicely validate the authors' hypothesis would imply to implant a human tumor selected among the MFNG Hi tumors, as in this context the anti-Jag1 antibody should show no effect on tumor growth, since Dll4/Notch signaling would be at work.

Answer: This is a very nice experiment that should involve several MFNG positive and negative tumors to definitively demonstrate the possibility pointed out by the reviewer. Because the suggested experiments are extremely time/money/animal consuming, we have addressed this issue using a

comparable in vitro assay with a set of MFNG negative and positive patient-derived samples from our bank of tumors. Our results that are shown in new figure S5B suggest that Jag1 and MFNG levels are in fact at the base of anti-Jag1 sensitivity. This information is mentioned in the text in page 11: “Using an in vitro system of patient-derived tumoroids (n=5), we confirmed that anti-Jag1 treatment was more effective in tumors carrying high levels of JAG1 and low MFNG such as PDOXT005, PDOXT007 and PDOXT008 (Figure S5B).”

Minor corrections:

The legend to Fig. 6 is incorrect. 6A=6B, 6B=6C, 6C=6D, 6D=6F, 6E=6F and 6F=6E. As there is no legend to 6A, it is unclear what the red staining represents in this panel.

Answer: We thank the reviewer for the observation. We have corrected this figure legend.

Comparable tumors should be shown in Fig. 2D.

Answer: We have changed the previous images by more comparable ones.

RT-PCR for the well-established direct Notch target gene Nrarp should be shown in Fig. 1D, as both c-Myc and Hes1 can be activated by Wnt signaling as well.

Answer: We have included this analysis not only in 1D but also in 3D.

There is no indication of the antibody used to detect Notch2 in Fig. S1F.

Answer: We have revised all information related with the antibodies used in the paper.

The images in Fig. 6F show an increased fibrosis or inflammation in the presence of the anti-Jag1 Ab, but not necessarily increased apoptosis. The decrease in Ki67+ cells is also not obvious in the images presented in the upper panel.

Answer: Based on the analysis of our pathologists, anti-JAG1 treatment increases the extent of tumor areas where transformed epithelial cells have

been essentially lost (likely by apoptosis or necrosis) and substituted by fibrotic infiltrate. Thus, and according to reviewer comments, we have changed the figure label that was confusing and substituted “apoptotic area” by “fibrotic area”, and also in the text in page 11 we specify: “IHC analysis of the tumors demonstrated that anti-Jag1 treatment imposed a significant increase in the extent of necrosis and fibrosis inside the tumor areas and a reduction in proliferation in the residual tumor mass as determined by ki67 staining (Figures 6E and 6F), which was not observed in the non-transformed adjacent colonic tissue (Figure 6G).”

In relation to the number of ki67 positive cells in either group, we performed automatic quantification of several areas and different tumors and this data is shown in 6F.

The relevant literature in the Introduction is not comprehensively cited.

Answer: We have revised it.

Reviewers' comments:

Reviewer #1 (Remarks to the Author):

Overall, the manuscript has been improved and the authors addressed the majority of the reviewer concerns. However, the following points need to be further clarified:

- Fig. 1G-H, the authors should revise the figure legend making it clear that the panels G and H show results regarding the same experiment.
- As supplementary Figures, the authors should add the IHC analysis of caspase-dependent cell death in Jag1lox/lox mouse model.
- In the Results section the authors claimed that "active-Caspase 3 and Alcian blue staining was colocalizing in cells delivered into the spheroid lumen". However, in the revised Fig. 3E the authors are showing a separate staining for Caspase 3 and for Alcian blue, respectively. Thus, the authors should revise their statement accordingly.
- The authors performed immunofluorescence analyses (IF) on paraffin-embedded sections referring to them as IHC analyses. Authors should revise the whole manuscript and Figures accordingly.
- The authors should include the legend describing the schematic model showed in Fig.7.

Reviewer #2 (Remarks to the Author):

The authors have improved their manuscript considerably - by improving the writing, removing inaccuracies, and by including new data.

They address the first of the three main concerns, by demonstrating that Dll4 cannot induce Notch1 signaling in the adenoma cells by dual IF stainings (Fig S2D)

Also, the analysis of Lgr5+ cells in the new Figure 2D is a clear improvement. However, the authors continue making very strong claims based on non-quantified data. Regarding the Lgr5 IF data, they write: "IHC analysis of LGR5 protein with a specific antibody confirmed these results, and excluded a defect in the normal intestinal stem cell compartment of Jag1 KO mice (Figure 2D)." Such a comment should not be made based on an IF image, but should be based on quantification of the Lgr5+ cell frequency on the images, or by FACS.

Unfortunately the attempts to provide more direct data on the role of MNFG for Jag1 dependent Notch-signaling remains correlative due to technical difficulties.

Minor:

The following sentence is difficult to understand:

"Paneth cell differentiation, negatively correlated with the number of functional Jag1 alleles, with KO tumors more frequently positive (70%) compared to HT (40%, $p < 0.05$) or WT (20%, $p < 0.01$) tumors (Figure 2E)."

Reviewer #3 (Remarks to the Author):

The authors have considerably improved their manuscript upon revision and they have addressed most of my questions. There are a few points that are still not very convincing (listed below), but overall the quality and significance of the results is correct.

1. Regarding the point of epithelial versus stromal function of Jag1, the authors now added a sentence suggesting that stromal Jag1 may also be important for tumor initiation or progression,

but they insist on keeping the phrasing "epithelial Jag1 is specifically required for tumor initiation"; the word "specifically" should be removed.

2. In Fig. S2D the authors observe "Treatment with 4-OH tamoxifen for 24-48 hours decreased ICN1 levels even in the presence of Dll4 (Figure S2D)". The ICN1 levels should disappear upon DAPT treatment, performed in Fig. 3E. The authors should show these data, as a comparison and control for specificity of the effect observed upon Jag1 deletion.

3. Several IHC in organoids, mainly in Fig. 2D and 3E, are still not convincing and incoherent between different figures:

- better quality images or higher magnifications should be provided for anti-Lgr5 staining, which looks mainly nuclear in Fig. 2D. Lgr5 should be localized at the membrane or in the cytoplasm, as it is observed in the anti-Lgr5 IF of Fig. S2B.
- the LYZ staining in Fig. 3E shows a higher green background in the 4-OHT and DAPT, as if the images in different panels had been taken at different exposure times. Anti-LYZ staining similar to the one shown in Fig. S2B would be clearer.
- also in Fig. 3E, the CAII levels seem decreased upon 4-OHT and DAPT, is this true?
- the Alcian blue staining in Fig. 3E appear very different in the 4-OHT and the DAPT panels: after 4-OHT, the blue coloration overlaps with cell nuclei? More importantly, no Alcian blue positive cell is visible upon DAPT treatment, as would be expected. The coloration in the organoid lumen is most likely due to non specific staining of apoptotic cells, shown in the cleaved Caspase3 panel. The authors should provide a Muc2 staining, as the one shown in Fig. S2B-C, which is definitely more specific to Goblet cells.
- in Fig. S2C no Ki67+ cells are observable in normal organoids, why?
- and in Fig. 4A, there is no ICN1+ cells in normal organoids?

4. I am not sure the new Fig. S5a is necessary and informative; it is very difficult to see any difference between control and anti-Jag1 treated animals, as one does not know what to look for.

5. In Fig. S5B the IHC anti-MFNG shows membrane staining in the first panel PDOXT001, but cytoplasmic and nuclear (?) staining in the remaining panels: what is the basis of this different localization?

Reviewer #1 (Remarks to the Author):

Overall, the manuscript has been improved and the authors addressed the majority of the reviewer concerns. However, the following points need to be further clarified:

Reviewer: Fig. 1G-H, the authors should revise the figure legend making it clear that the panels G and H show results regarding the same experiment.

Answer: We have clarified in the legend that results in F-G and in H correspond to different experiments.

Reviewer: As supplementary Figures, the authors should add the IHC analysis of caspase-dependent cell death in Jag1lox/lox mouse model.

Answer: We did not detect any increase in the number of cleaved-Caspase3 positive cells neither in the non-transformed crypts neither in the remaining tumor of the Jag1lox/lox mice compared with the other genotypes, however we are including this information as supplementary data in page 5: “we did not detect a significant increased in the number of cleaved-caspase3 positive cells (apoptotic) in any Jag1 genotype (Figure S1G).”

And in page 6, we also mention: “Together, these results indicate that Jag1 is not essential for normal intestinal homeostasis, but it is required for β -catenin-driven adenoma formation. Thus, genetic deletion of epithelial Jag1 or systemic inhibition using a blocking antibody is sufficient to prevent adenoma formation in the $Apc^{Min/+}$ mice, with no evident sign of toxicity.”

Reviewer: In the Results section the authors claimed “active-Caspase 3 and Alcian blue staining was colocalizing in cells delivered into the spheroid lumen”. However, in the revised Fig. 3E the authors are showing a separate staining for Caspase 3 and for Alcian blue, respectively. Thus, the authors should revise their statement accordingly.

Answer: The reviewer is correct with this sentence is misleading, thus in page 8 we have the text to: “active-Caspase 3 and Alcian blue stainings were both detected in cells delivered into the spheroid lumen suggesting that Notch inactivation following Jag1 deletion imposes the terminal mucosecretory differentiation of adenoma cells leading to cellular apoptosis.”

Reviewer: The authors performed immunofluorescence analyses (IF) on paraffin-embedded sections referring to them as IHC analyses. Authors should revise the

whole manuscript and Figures accordingly.

Answer: Thank you for the observation, we have revised the manuscript accordingly.

Reviewer: The authors should include the legend describing the schematic model showed in Fig.7.

Answer: We have included this legend that was inadvertently omitted.

Reviewer #2 (Remarks to the Author):

Reviewer: The authors have improved their manuscript considerably - by improving the writing, removing inaccuracies, and by including new data.

Answer: We thank the reviewer for this comment.

Reviewer: They address the first of the three main concerns, by demonstrating that Dll4 cannot induce Notch1 signaling in the adenoma cells by dual IF stainings (Fig S2D)

Answer: This is correct.

Reviewer: Analysis of Lgr5+ cells in the new Figure 2D is a clear improvement. However, the authors continue making very strong claims based on non-quantified data. Regarding the Lgr5 IF data, they write: "IHC analysis of LGR5 protein with a specific antibody confirmed these results, and excluded a defect in the normal intestinal stem cell compartment of Jag1 KO mice (Figure 2D)." Such a comment should not be made based on an IF image, but should be based on quantification of the Lgr5+ cell frequency on the images, or by FACS.

Answer: We have quantified the number of LGR5+ cells in the normal crypt and the tumors (from 15 fields per condition counted), and new data is included in Figure 2D and in the text in pages 6-7: "IHC analysis of LGR5 protein with a specific antibody confirmed a reduction of the stem cell-like tumor population in the Jag1 KO mice (from 48% in the WT to 21% in average) that did not affect the normal intestinal stem cell compartment of the crypts (Figure 2D)."

Unfortunately the attempts to provide more direct data on the role of MNFG for Jag1 dependent Notch-signaling remains correlative due to technical difficulties.

Answer: We agree with the reviewer comment and we are planning to address this issue in

subsequent projects.

Minor:

The following sentence is difficult to understand:

"Paneth cell differentiation, negatively correlated with the number of functional Jag1 alleles, with KO tumors more frequently positive (70%) compared to HT (40%, $p < 0.05$) or WT (20%, $p < 0.01$) tumors (Figure 2E)."

Answer: We have change this sentence in page 7 to: "presence of Paneth cell in the tumors, as determined by lysozyme (Lyz1) detection, negatively correlated with the number of functional Jag1 alleles, being KO tumors more frequently Lyz1 positive (70%) compared to HT (40%, $p < 0.05$) or WT (20%, $p < 0.01$) tumors (Figure 2E)."

Reviewer #3 (Remarks to the Author):

The authors have considerably improved their manuscript upon revision and they have addressed most of my questions. There are a few points that are still not very convincing (listed below), but overall the quality and significance of the results is correct.

Answer: We thank the reviewer for this comment but also for his/her suggestions that have already contributed to improve the quality of our work.

Reviewer (1): Regarding the point of epithelial versus stromal function of Jag1, the authors now added a sentence suggesting that stromal Jag1 may also be important for tumor initiation or progression, but they insist on keeping the phrasing "epithelial Jag1 is specifically required for tumor initiation"; the word "specifically" should be removed.

Answer: We have removed it.

Reviewer (2): In Fig. S2D the authors observe "Treatment with 4-OH tamoxifen for 24-48 hours decreased ICN1 levels even in the presence of DII4 (Figure S2D)". The ICN1 levels should disappear upon DAPT treatment, performed in Fig. 3E. The authors should show these data, as a comparison and control for specificity of the effect observed upon Jag1 deletion.

Answer: We have included this control.

Reviewer (3): Several IHC in organoids, mainly in Fig. 2D and 3E, are still not

convincing and incoherent between different figures: better quality images or higher magnifications should be provided for anti-Lgr5 staining, which looks mainly nuclear in Fig. 2D. Lgr5 should be localized at the membrane or in the cytoplasm, as it is observed in the anti-Lgr5 IF of Fig. S2B.

Answer: We were also surprised to see that Lgr5 staining was not restricted to the membrane as we expected based on Lgr5 structure and function, and comparable to what we previously observed in the whole mount staining of 3D growing spheroids. However, we found that this “expected” distribution was primarily lost in the paraffin sections of the intestine and changed into a cytoplasmic plus nuclear staining, but maintaining the cellular distribution that is restricted to few cells at the bottom of the intestinal crypts (now quantified in new Figure 2D). We have confirmed this pattern and the cellular specificity of the staining with a second anti-Lgr5 antibody from Abcam (Ab219107). Following the reviewer suggestion we are including a detail of the staining. Unfortunately, we did not find in the literature any endogenous Lgr5 staining of intestinal crypts with the expected membranous staining pattern.

Reviewer (4): the LYZ staining in Fig. 3E shows a higher green background in the 4-OHT and DAPT, as if the images in different panels had been taken at different exposure times. Anti-LYZ staining similar to the one shown in Fig. S2B would be clearer.

Answer: Images in the panels corresponding to the same staining were all taken in the same conditions, otherwise any comparison would be impossible. In relation to the differences between staining in Figure 3E and S2B, 3E correspond to 2.5-micron paraffin sections, which results in a much lower signal (in general) than that obtained in the whole mount staining. However, we consider that even when intensity of the signal is reduced the quality of the images in the paraffin sections is much better.

Reviewer (5): also in Fig. 3E, the CAII levels seem decreased upon 4-OHT and DAPT, is this true?

Answer: Yes, in all spheroids we analyzed, we detected a slight decrease in the CAII levels in both 4-OHT and DAPT but since IF is not a quantitative method and the differences are subtle, we are not convinced about the functional relevance of the observation. However, since the possibility that Notch inhibition reduces CAII levels exist, we have change the text to: “4-OH tamoxifen treatment considerably diminished the number of proliferating Ki67-positive cells and increased the quantity of lysozyme-(Paneth) and Alcian Blue-positive (mucosecretory) cells without significantly affecting CAII

levels.”

Reviewer (6): the Alcian blue staining in Fig. 3E appear very different in the 4-OHT and the DAPT panels: after 4-OHT, the blue coloration overlaps with cell nuclei? More importantly, no Alcian blue positive cell is visible upon DAPT treatment, as would be expected. The coloration in the organoid lumen is most likely due to non-specific staining of apoptotic cells, shown in the cleaved Caspase3 panel. The authors should provide a Muc2 staining, as the one shown in Fig. S2B-C, which is definitely more specific to Goblet cells.

Answer: The reviewer is correct that nuclear Alcian blue staining in the 4-OHT treated cells could represent some type of artifact, although artifactual staining tend to be more general and not restricted to a particular condition, and would not affect our conclusions. About the possibility that apoptotic cells might be non-specifically stained with Alcian Blue is a plausible explanation, which can be applied to any other staining, however our interpretation is that mature goblet cells (the ones that are high Alcian Blue positive) are rapidly released to the lumen of the spheres where they enter into the apoptotic pathway. About the possibility to include MUC2 staining in Fig. 3E, we have tested this staining in our remaining paraffin sections and we did not feel that the results are different or better than the ones obtained with Alcian Blue. So we decided to maintain Alcian Blue in the figures but include MUC2 staining here for the reviewer.

Reviewer (7): in Fig. S2C no Ki67+ cells are observable in normal organoids, why?

Answer: We apologize as we agree with the reviewer that the intensity of the ki67 staining in these organoids was very low. We have now changed the original image by a better-quality one. However, we have to admit that the quality of the whole-mount staining of the organoids is variable and not always as we would like.

Reviewer (8): and in Fig. 4A, there is no ICN1+ cells in normal organoids?

Answer: We have carefully revised the above-mentioned images and we consistently detected some ICN1 staining in the normal organoids although the signal was much less intense than the staining detected in the few spheroids cells showing a solid nuclear staining pattern. Instead the staining in the organoids but also in most of the spheroid cells is a spotted pattern in the nucleus.

Reviewer (9): I am not sure the new Fig. S5a is necessary and informative; it is very difficult to see any difference between control and anti-Jag1 treated animals, as one does not know what to look for.

Answer: We agree with the reviewer that these images are not very informative but we have included them following the suggestion of reviewer 1.

Reviewer (10): In Fig. S5B the IHC anti-MFNG shows membrane staining in the first panel PDOXT001, but cytoplasmic and nuclear (?) staining in the remaining panels: what is the basis of this different localization?

Answer: Our interpretation is that in the absence of a high cytoplasmic-membranous staining, which is the case of most human tumoroids, the antibody was detecting a more diffuse cytoplasmic staining and some nuclear staining in some cells (not all of them). However, we do not think that this nuclear staining is specific and in the case it was, this nuclear protein should not be functional in terms of Notch1 glycosylation.

REVIEWERS' COMMENTS:

Reviewer #2 (Remarks to the Author):

The authors have continued improving the paper. They have addressed the first of my remaining concerns by quantifying the frequency of the Lgr5+ cells in the tumors and normal tissue of Jag1+/+ and Jag1flox/flox mice. This data supports their conclusions on the role of Jag1 in adenoma formation. My other concern - the so far only correlative evidence on the role of MNFG for Jag1 dependent Notch-signaling - remains, but it can be argued to go beyond the scope of this manuscript.

Reviewer #3 (Remarks to the Author):

The authors have satisfactorily addressed my previous comments and suggestions. The fact that the authors themselves state "the quality of the whole-mount staining of the organoids is variable and not always as we would like" still remains a bit worrisome, as many conclusions of this paper are drawn based on such organoid staining experiments, which must be reliable. However, the general message of a switch in ligand availability in normal stem cells versus tumours is interesting and warrants publication in my view.

Reviewer #2 (Remarks to the Author):

The authors have continued improving the paper. They have addressed the first of my remaining concerns by quantifying the frequency of the Lgr5+ cells in the tumors and normal tissue of Jag1^{+/+} and Jag1^{flox/flox} mice. This data supports their conclusions on the role of Jag1 in adenoma formation. My other concern - the so far only correlative evidence on the role of MNFG for Jag1 dependent Notch-signaling - remains, but it can be argued to go beyond the scope of this manuscript.

Reviewer #3 (Remarks to the Author):

The authors have satisfactorily addressed my previous comments and suggestions.

The fact that the authors themselves state “the quality of the whole-mount staining of the organoids is variable and not always as we would like” still remains a bit worrisome, as many conclusions of this paper are drawn based on such organoid staining experiments, which must be reliable. However, the general message of a switch in ligand availability in normal stem cells versus tumours is interesting and warrants publication in my view.

Answer: We thank the reviewers for their comments.